# The admixed brushtail possum genome reveals invasion history in New Zealand and novel imprinted genes

Donna M. Bond [1,17], Oscar Ortega-Recalde[1,17], Melanie K. Laird[1,17], Takashi Hayakawa [2], Kyle S. Richardson[1,15], Finlay.C. B. Reese [1], Bruce Kyle[1], Brooke E. McIsaac-Williams[1], Bruce C. Robertson [3], Yolanda van Heezik[3], Amy L. Adams[3], Wei-Shan Chang [4,16], Bettina Haase[5], Jacquelyn Mountcastle[5], Maximilian Driller[6], Joanna Collins[7], Kerstin Howe [7], Yasuhiro Go [8,9,10], Francoise Thibaud-Nissen [11], Nicholas C. Lister[12], Paul D. Waters [12], Olivier Fedrigo[5], Erich D. Jarvis[5,13,14], Neil J. Gemmell [1], Alana Alexander [1] & Timothy A. Hore [1] ✉

Combining genome assembly with population and functional genomics can provide valuable insights to development and evolution, as well as tools for species management. Here, we present a chromosome-level genome assembly of the common brushtail possum (*Trichosurus vulpecula*), a model marsupial threatened in parts of their native range in Australia, but also a major introduced pest in New Zealand. Functional genomics reveals post-natal activation of chemosensory and metabolic genes, reflecting unique adaptations to altricial birth and delayed weaning, a hallmark of marsupial development. Nuclear and mitochondrial analyses trace New Zealand possums to distinct Australian subspecies, which have subsequently hybridised. This admixture allowed phasing of parental alleles genome-wide, ultimately revealing at least four genes with imprinted, parent-specific expression not yet detected in other species (*MLH1*, *EPM2AIP1*, *UBP1* and *GPX7*). We find that reprogramming of possum germline imprints, and the wider epigenome, is similar to eutherian mammals except onset occurs after birth. Together, this work is useful for genetic-based control and conservation of possums, and contributes to understanding of the evolution of novel mammalian epigenetic traits.

The common brushtail possum (*Trichosurus vulpecula*) is a nocturnal arboreal marsupial native to Australia. Like other marsupials, the brushtail possum gives birth to altricial young—gestation lasts only 17.5 days. Possums at birth are hairless, without overt sexual differentiation and do not even possess full respiration capacity in the lungs[1]. The first 100 days of life in the pouch are spent moving very little and with eyes shut, completing organogenesis and growth while continuously suckling from a teat[2,3] (Fig. 1a). As such, it could be argued that the most major developmental transition for possums is at weaning and

exit from the pouch around 6 months of age. Despite this, physiological changes associated with extended lactation and weaning have not been well studied, particularly in a genome-wide manner.

In addition to studies of their unique adaptations, marsupials are often used as a model for comparative evolutionary studies[4–6]. Of these, one of the most interesting is genomic imprinting—a form of monoallelic gene expression that is dependent on the sex of the parent from which an allele is inherited[7,8]. There are approximately 100–200 imprinted genes in humans and mice, characterised by paternal-

specific expression of factors enhancing growth, and maternal-specific expression of those suppressing growth[9]. The most popular theory explaining why these genes would forgo the benefits of diploidy is known as the kinship, or parental-conflict, hypothesis[10,11]. The kinship hypothesis predicts genes with paternal-specific expression will maximise the use of maternal resources and increase the fitness of fathers. In contrast, maternally-derived genes act to control growth such that total reproductive output is maximised, irrespective of who the father is.

Marsupials like possum provide an interesting test for the parental-conflict hypothesis because there is limited opportunity for paternally-derived genes to influence maternal resource provision within the short-lived placenta[7,12]. Indeed, most of the genes that are imprinted in humans and mice appear to be not imprinted, or are without orthologues, in marsupials[7,13–15]; however, genome-wide searches for marsupial imprinted genes are limited[16,17]. Other than the genes controlling imprinted X-chromosome inactivation[4], genomic imprinting is yet to be studied in brushtail possums.

In addition to their value as a model for marsupial development and understanding mammalian evolution, possums are an iconic and protected species in Australia. For Aboriginal Australians, possums represent a cultural treasure, whereby cured pelts are used as a canvas

upon which individual life-stories are depicted[18]. Possums can thrive in urban areas and are thus the native mammals most likely to come into human contact. Consequently, possums are a focus of relocation and rehabilitation efforts in areas where they are threatened[19–21]. These conservation efforts are hampered, at least to some extent, by the lack of genomic resources[19].

In striking contrast to Australia, possums have destroyed many native forest ecosystems in Aotearoa New Zealand, following their first importation in 1858[22]. Ecological damage occurs mainly through defoliation of trees[23], but also through opportunistic invertebrate and bird predation[24]. Considering their additional role as a reservoir for bovine tuberculosis[25], possums are often viewed as the most significant vertebrate pest species in New Zealand[26]. Efforts to control and locally eradicate possums have focussed on poisoning, trapping and shooting; however, more humane and effective approaches involving suppression of possum fertility have been hampered by a lack of genetic information about the species.

To address this, we have produced a near-complete possum genome assembly as part of the Vertebrate Genomes Project, and undertaken a comprehensive functional genomics study. In addition to providing a basic understanding of marsupial pouch young development, we have identified novel communication genes, and imprinted

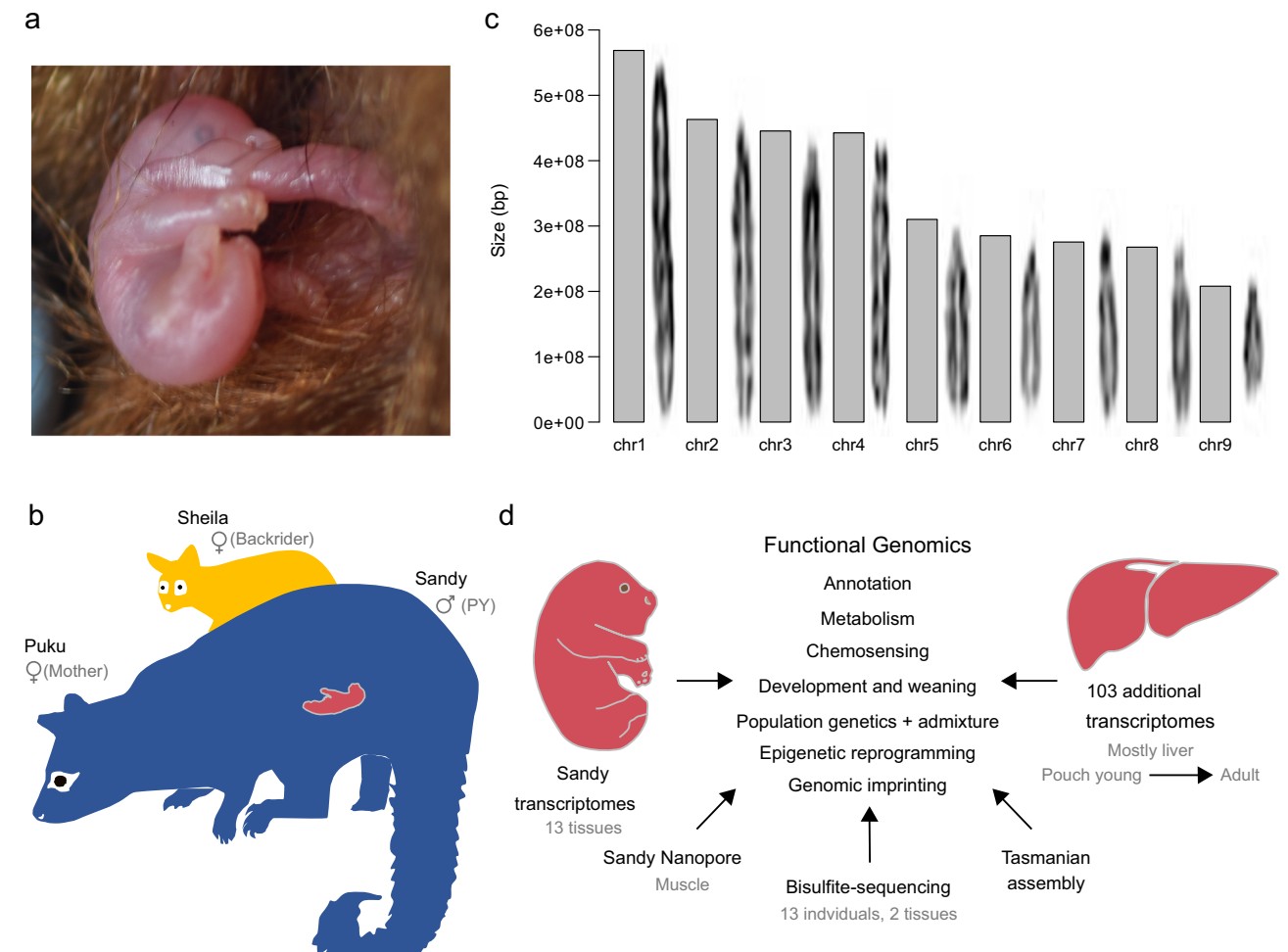

**Fig. 1 | Chromosomal assembly of the possum genome. a** Possum pouch young are altricial and complete the majority of development suckling inside the pouch. **b** The individual sequenced (Sandy) was a male pouch young (PY), collected alongside his mother (Puku) and juvenile 'backrider' sister (Sheila), who had not long left the pouch. **c** The assembled genome from Sandy features 99.88% of the sequence assigned to chromosomes, matching the karyotype[28]. **d** A range of

functional genomics data supports the assembly, including RNA-sequencing (mostly liver) from a wide population of possums and 13 tissues from Sandy. Also performed was DNA methylation profiling, featuring nanopore and bisulfite sequencing for genomic imprinting and germline reprogramming analysis, respectively, and a sequence assembly of a Tasmanian possum.

loci not yet discovered in other species. By quantifying admixture in New Zealand possums, and tracing diversity to their source populations in Australia, we aid both pest management in New Zealand and conservation in their native range.

## Results

### Chromosomal assembly of the possum genome

The main brushtail possum genome assembly was generated from the forearm muscle of a 22-day-old male pouch young we called 'Sandy' (Fig. 1b). In addition to Sandy, tissue was collected from his sister (Sheila), a 'backrider' of approximately 180 days old, and mother (Puku) following trapping. This trio and many other possum samples we collected were from the Dunedin region in southern New Zealand—we concentrated our efforts here on account of it featuring multiple early possum introductions, both from the first site of establishment in nearby Southland, but also directly from Tasmania and mainland Australia[22].

We generated the assembly following the Vertebrate Genomes Project pipeline v1.5[27], including haplotype phased contigs from Pac-Bio long-read sequencing, polishing nucleotide errors with short-reads, scaffolding the contigs into chromosomes with 10x Genomics linked reads, Bionano Genomics optical maps, Arima Genomics Hi-C chromatin maps, and curation for structural errors and naming of chromosomes (GenBank accession GCA_011100635.1). Almost all sequence (99.88%) mapped to chromosomes, matching the karyotype[28] (Fig. 1c), and possessing strong assembly rates (scaffold N50, 442,560,073 bp; contig N50, 4,314,688 bp). Gene annotation was assisted by long-read IsoSeq RNA-sequencing from muscle, brain and testis tissue (GenBank accession: GCF_011100635.1). Our assembly registered a BUSCO score of 96.8% complete genes, better than the most commonly used marsupial genomes (e.g., *Monodelphis domestica*, 91.6% complete)[29], with the exception of the koala (98% complete)[30,31] (Supplementary Table 1). Due to hemizygosity, the X-chromosome suffered increased fragmentation relative to the autosomes. Nevertheless, we assembled approximately ~1.28 Mbp of Y-chromosome sequence from 3 separate scaffolds that were specific to males (Fig. S1). At least one gene on these scaffolds, synaptonemal complex element protein 1 (*SYCE1Y*), is a novel Y-linked gene in mammals with differential expression compared to its X-linked homologue, *SYCE1X*.

### RNA sequencing reveals unique physiological adaptations associated with altricial birth, extended lactation and weaning

To assist with functional annotation of the possum genome, we undertook RNA-sequencing from 13 of Sandy's tissues. We also produced a further 103 transcriptome reactions from tissues (mostly liver) collected over a range of developmental stages and locations in the Dunedin region (southern New Zealand; Fig. 1d). Many genes upregulated in possum pouch young relative to adults appeared characteristic of fetal life in eutherian mammals. For example, alpha-fetoprotein (*AFP*; *LOC118852573*) is the major albumin protein during fetal life in humans and mice and its expression is rapidly silenced after birth[32]. Yet we saw high expression in possum pouch young, including juvenile pouch young >120 days post birth (Fig. S2a). Silencing of globin gene expression[33] was also observed in early pouch young, thus reflecting the transfer of blood system production from the liver to bone marrow in post-natal marsupials[34] (Fig. S2b; Supplementary Data 1).

Perhaps the most striking gene expression change we observed in the liver between pouch young and adults was associated with the metabolic effects of weaning (Fig. 2a; Supplementary Data 2). Genes associated with the Leloir pathway, involved with the catabolism of galactose and other carbohydrates, were high in pouch young, but low in adults. In contrast, cytochrome P450-family (*CYP*) genes associated with the first major stage of metabolic breakdown (i.e., oxidation) were

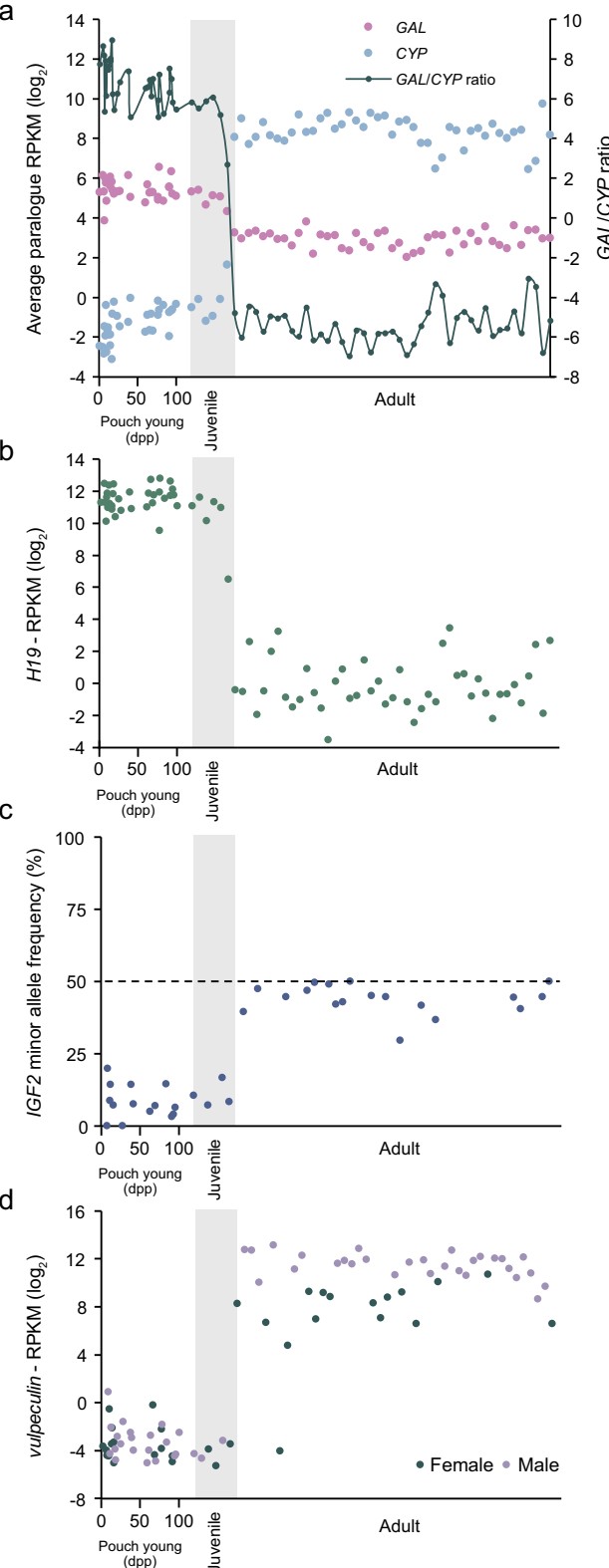

**Fig. 2 | Gene expression changes associated with weaning. a** Galactose metabolism (*GAL*, pink), cytochrome P450 (*CYP*, blue) paralogues (left) and the *GAL/CYP* ratio (green; right), **b** *H19*, **c** *IGF2* minor allele and **d** *vulpeculin* (females=green, males=purple) expression changes in liver RNA-sequencing from pouch young and adult possums. dpp, days postpartum. Source data are provided as a Source Data file.

low in pouch young but became highly expressed in adults. These gene expression changes reflect liver maturation and a metabolic transition to processing a range of dietary molecules at weaning. Interestingly, we found that the 'back-rider' Sheila (Sandy's half-sister), had a low galactase/CYP ratio, indicative of weaning. This result provides molecular proof that at least some backriders stay with their mothers post-weaning, perhaps to learn browsing techniques.

To uncover molecular signals of unique marsupial traits associated with weaning, we searched for genes associated with the galactase/CYP expression ratio. The gene with the highest correlation to the galactase/CYP ratio encoded the imprinted long non-coding RNA, *H19* (*LOC118853838*; Fig. 2b). *H19* is thought to help orchestrate the paternal-specific expression of *insulin-like growth factor 2* (*IGF2*) in mice[35], an iconic and intensively studied imprinted gene. Similar regulation of *IGF2/H19* has been proposed for the tammar wallaby, an Australian macropodid marsupial[36,37]. To explore this further in possum, we searched for single-nucleotide polymorphisms (SNPs) within *H19* and *IGF2* of Sandy and other individuals we collected, and tested their expression. We found exclusive maternal expression of *H19*, whereas *IGF2* was paternally expressed in pouch young and biallelically expressed in adults where *H19* expression was silenced (Fig. 2b, c; Fig. S2c).

Genes anti-correlated with the galactase/CYP weaning signature were also identified. Two of the most anti-associated were lipocalin family members (*LOC118842298* and *LOC118843263*), known for binding and transporting small hydrophobic molecules such as steroid hormones, odorants, retinoids, and lipids[38–40]. Homology searches revealed that *LOC118842298* encoded vulpeculin, previously known in possum only by its protein sequence[41,42], with the other being a close relative (which we have named ganderin), situated amongst a mini-cluster of lipocalin genes on chromosome 3 (Supplementary Table 2). Vulpeculin is notable as being the most abundant protein in possum urine[42], implying it acts in a similar manner to a large family of related lipocalins, the major urinary proteins in rodents, which are expressed in the liver but excreted in urine and heavily implicated in pheromone biology and chemical communication[39]. Interestingly, we found that expression of *vulpeculin* and *ganderin* is somewhat variable in females, but is consistent and much higher in males (Figs. 2d, S2d).

### Early invasion history and admixture of possums in New Zealand

Brushtail possums are widely distributed within their Australian home range, with up to 6 recognised subspecies[43]. Of these, *Trichosurus vulpecula fuliginosus* from lutruwita Tasmania, and *Trichosurus vulpecula vulpecula* from south-east Australia, have well-documented importation into New Zealand[22]. The extent to which these subspecies are hybridising has not yet been fully described—elsewhere in New Zealand, certain populations maintain significant substructure, whereas others are freely interbreeding[44–48]. In all cases, genetic tracing to reveal the source Australian populations remains incomplete.

We hypothesised that possums from the Dunedin area may hold considerable genetic variation as this was an early and intensive site of possum release[22]. Indeed, mitochondrial sequencing from 3 Dunedin locations revealed 4 distinct mitochondrial haplotypes perhaps representing separate Australian source populations. One of these haplotypes clustered with possum samples originating near Sydney in New South Wales (NSW), with another closely related to the Tasmanian reference possum we sequenced (Fig. 3a). The other two haplotypes aligned to D-loop sequence from possums sourced from separate mainland populations (99.1% and 97.0% identity, respectively)[48]. We assessed nuclear genetic diversity from our RNA-sequencing data and reference samples[49,50], finding standardised multilocus heterozygosity (sMLH) is higher on average in Dunedin possums than in the native Australian range, even when restricting comparisons to possums sharing the same mitochondrial haplotype. When compared to RNA-sequencing data from Australian mainland possums[50], and low-

coverage genomic sequencing of the Tasmanian individual, we found that diversity among Dunedin possums appears to be driven by admixture between diverse sources in Tasmania and mainland Australia (Fig. 3b).

### Identification of novel marsupial imprinted genes

Relative to eutherian mammals, marsupial placentation affords less opportunity for manipulation of maternal physiology. As such, it is assumed that genomic imprinting is relatively reduced in marsupials, a proposition supported by biallelic expression or absence of many eutherian imprinted genes[7,14–17,51]. Nevertheless, the possibility that marsupials possess a novel set of imprinted genes that relate to their unique biology has not been tested outside of *M. domestica*[16,17].

To perform an effective imprinted gene search, the ability to distinguish parental alleles over a large number of genes is essential. High levels of heterozygosity in Sandy and the other admixed New Zealand possums (Fig. 3a) provided a unique opportunity to search for novel marsupial imprinted genes. Initially, we tried searching for allele-specific expression in RNA-sequencing datasets, however, this resulted in a large number of candidate genes, many of which were false positives involving pseudogenes and repetitive regions[52]. Consequently, we altered our discovery pipeline to first identify allele-specific methylation (ASM) at candidate imprinting control regions[53–55] using long-read nanopore sequencing (Fig. S3a). We started with an initial list of 173 sites displaying ASM (Supplementary Data 3), which included known marsupial imprint control regions (ICRs) such as intron 12 of *IGF2R*[56] (Fig. S3b) and the *PEG10* promoter[57] (Fig. S3c). We then searched for monoallelic expression of SNPs expressed in nearby genes. As heterozygous SNPs were so clearly identified in Sandy's genome due to 60-fold sequencing coverage, we started with an analysis of his transcriptome set.

In addition to known marsupial imprinted genes such as *IGF2R* and *PEG10*, we found 3 sites of ASM (Fig. 4a–c) associated with monoallelic expression of adjacent or overlapping genes in all of Sandy's tissues (Fig. 5a–d; Fig. S4a). This included the *MLH1* and *EPM2AIP1* genes linked to a single intervening region of ASM, as well as single gene loci featuring *UBP1* and *GPX7* (Fig. 5a–d).

To further test the monoallelic nature of these genes, we then examined our RNA-sequencing datasets from other individuals. Although significant allelic variation was found in these genes, no single individual appeared to express both alleles—a violation of the Hardy-Weinberg equilibrium (Supplementary Data 4). Amplicon sequencing was then used to definitively prove which individuals were heterozygous despite showing expression of only one allele (Fig. 5e–h; Fig. S4b–e; Supplementary Data 4).

While monoallelic expression was clear for *MLH1/EPM2AIP1*, there appeared to be biased expression of *GPX7* and *UBP1*. Moreover, our experiments up to this point had not definitively shown allelic dependence on parental sex. To address this, we genotyped the mothers of heterozygous offspring at our loci of interest—in instances where a mother was homozygous, this allowed us to assign the parental origin of alleles. In every case, we confirmed that *GPX7* was expressed from the maternal allele ($n = 7$), whereas *MLH1* ($n = 5$), *EPM2AIP1* ($n = 5$), and *UBP1* ($n = 4$), were all expressed from the paternal allele (Table 1; Supplementary Data 4).

### Global methylation erasure in primordial germ cells coincides with the reprogramming of imprint control regions

In humans and mice, epigenetic marks acquired in one generation are generally not passed on to the next because of genome-wide epigenetic erasure in the germline[58,59]. By far the most distinctive methylation erasure event occurs in primordial germ cells (PGCs), in which high starting levels of methylation (70-80%) are removed through active and passive means, leaving only 5% methylation at its lowest point[60,61]. In contrast, while there is an alteration of methylation

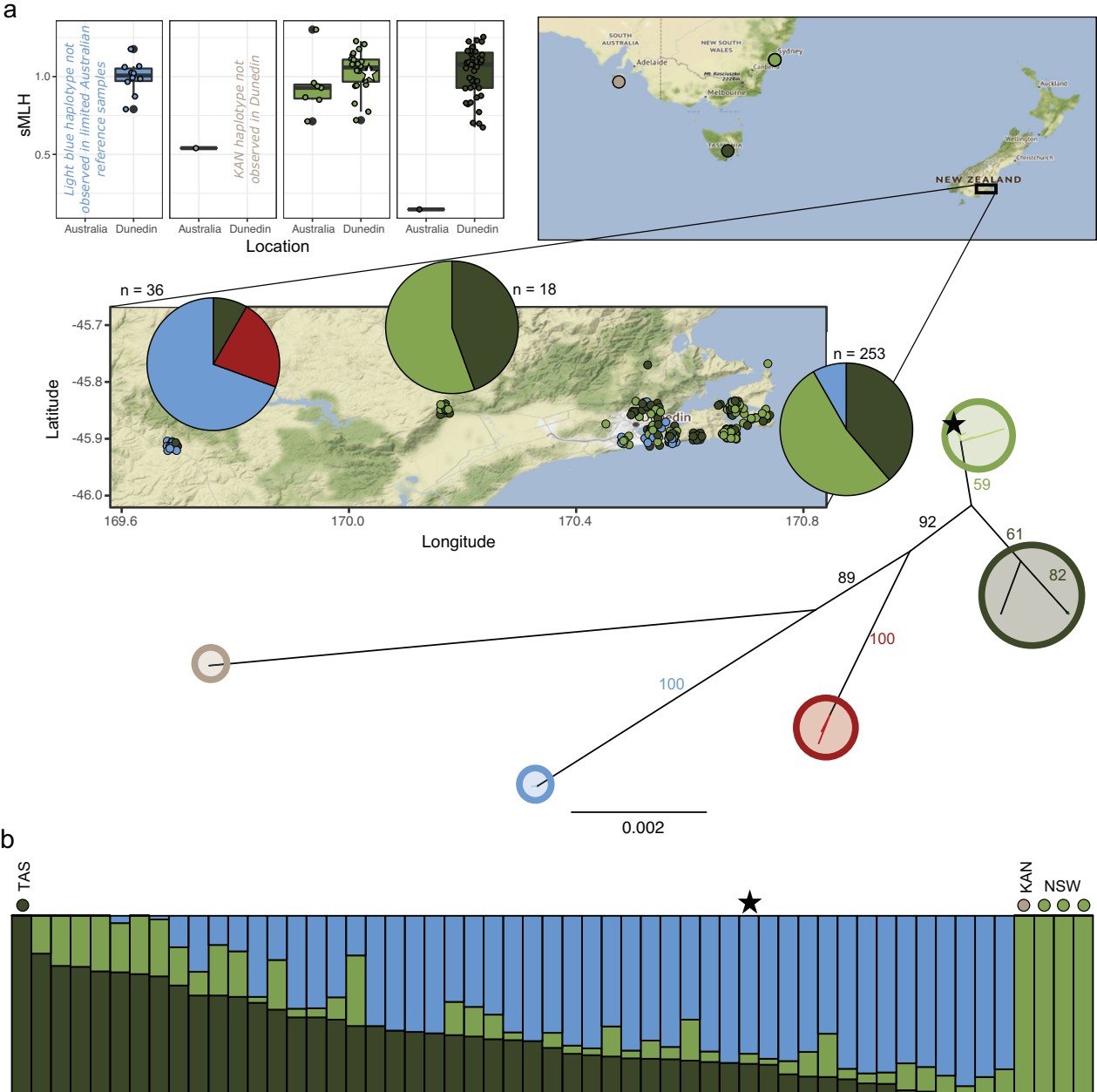

**Fig. 3 | Diverse geographic origins of admixed Dunedin possums lead to high genetic diversity. a** Standardised multilocus heterozygosity (sMLH, upper left) of Australian and Dunedin possums, obtained from RNA-sequencing variant calls (biologically independent animals, $n = 10, 1, 7, 25, 1$ and 45, left to right, respectively). Boxplot minima and maxima (large black dots) indicate points beyond whiskers. Whiskers are 1.5*interquartile range above and below the 75th percentile and 25th percentile (upper and lower whisker respectively), which form the bounds of the box, with the centre of the box representing the median. Clades were obtained from the maximum likelihood phylogenetic reconstruction of mitogenomes, utilising reference samples from New South Wales (NSW, light green), Kangaroo Island (KAN, beige), and Tasmania (TAS, dark green). Haplotypes from NSW and TAS, but not KAN, were present in Dunedin samples, as well as two likely mainland Australian haplotypes, identified by homology to D-loop sequence[48] (red and blue). **b** Nuclear admixture analysis for Dunedin samples relative to reference samples from TAS (dark green) and NSW/KAN (light green), as well as another likely mainland source (blue). Results for Sandy, the reference genome individual, are indicated (star). Source data are provided as a Source Data file.

patterns in the germline of non-mammalian vertebrates like zebrafish, global methylation erasure in PGCs apparently does not occur[62–66]. This provides for the interesting possibility that acquired traits underpinned by DNA methylation can be inherited in at least some vertebrate groups[58,67,68].

As a divergent mammalian group, methylation dynamics in the marsupial germline are somewhat unclear. Global loss of PGC methylation has been observed in wallaby[69], however, the immunofluorescence techniques used were not quantitative. To explore genome-wide methylation changes in possum development, we

isolated PGCs from the gonads of male pouch young up to 106 days of age (Fig. 6a). Low-coverage bisulfite sequencing was used to estimate global DNA methylation levels throughout post-natal development. Methylation levels were high immediately after birth (69.7%), but dropped to 41.1% by 10 days postpartum (dpp) (Fig. 6b; Supplementary Data 5). Subsequently, remethylation occurred such that at 25 dpp, 60-70% methylation was observed ($t$-test, $p = 0.001$), consistent with adult sperm.

To establish if the novel imprinting control regions we identified are reprogrammed during this period of methylation erasure, we

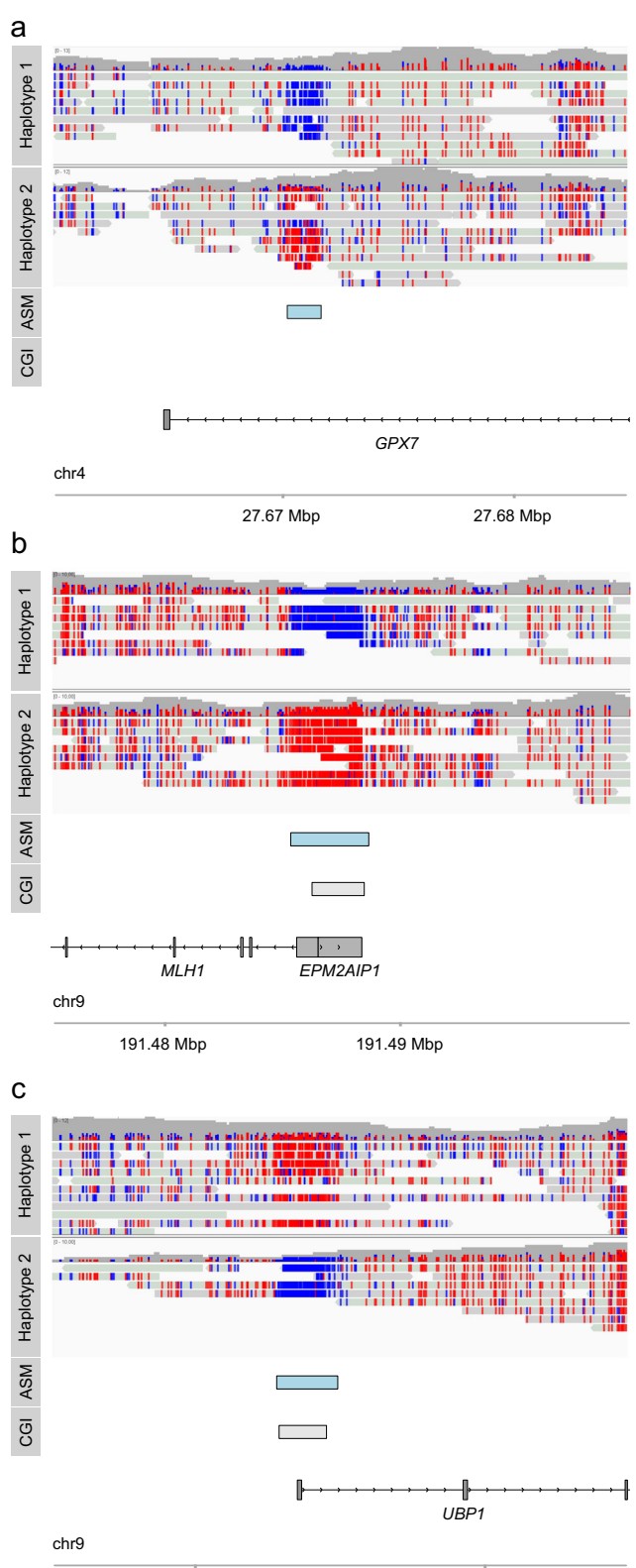

**Fig. 4 | Novel allele-specific methylation sites identified by nanopore sequencing.** Methylation haplotypes associated with three selected loci: *GPX7* (**a**), *MLH1/EPM2AIP1* (**b**) and *UBP1* (**c**). Methylated cytosines are red, unmethylated cytosines are blue; ASM allele-specific methylation, CGI CpG island.

undertook deep sequencing of two libraries (13 and 16 dpp), and compared these to libraries of control somatic cells from the same individuals (i.e. non-germline gonadal cells) and adult sperm (Fig. 6c). From this analysis, it was clear that the control somatic cells were 50% methylated over the regions we found that had ASM in Sandy (Fig. 4a–c; Fig. S3b, c and Fig. S5b). In contrast, PGC methylation in the same regions was 5.06-20.7%, implying that the erasure of methylation imprints in the germline coincides with global DNA demethylation. Interestingly, sperm methylation at the ASM sites was even lower than PGCs for *IGF2R*, *PEG10*, *MLH1* and *UBP1* (1.39–1.80%); however, *H19* and *GPX7*, were fully methylated (94.24–95.79%) (Fig. 6c). This supports our findings of paternal- and maternal-specific expression for these genes, respectively (Fig. 5a–h; Fig. S4b–e; Supplementary Data 4), and implies the ASMs we discovered to act as germline imprinting control regions (ICRs). Moreover, our data for *PEG10* and *IGF2R* is consistent with tammar wallaby[56,57].

To identify other putative ICRs with methylation patterns indicative of imprinting, we assessed methylation levels of the remaining ASM sites we identified in the PGC, somatic cell and adult sperm datasets (Fig. 6d, top panel; excluding those ASM sites with missing data, *n* = 32). We found 41 ASM sites had somatic methylation levels that fall within the range exhibited by germline ICRs (30–57%). Of these, 9 ASM sites were associated with (+/– 1 Mb distance) 13 genes displaying monoallelic expression in at least Sandy's brain, liver or muscle (Fig. 6d, bottom panel; Supplementary Data 3). While not as robustly tested as *MLH1/EPM2AIP1*, *GPX7* and *UBP1*, we nonetheless consider these ASM sites and genes to be strong imprinted gene candidates worthy of further testing.

## Discussion

The brushtail possum is protected across their native range in Australia but is considered a noxious pest in New Zealand. Efforts to manage their respective populations are simultaneously undermined by a lack of genetic information. We have addressed this by producing a chromosome-scale assembly of the possum genome that is amongst the best for any marsupial to date, due at least in part to the incorporation of long-read sequencing and haplotype phasing[27].

The value of this reference assembly sequence is amplified through complementary functional genomics obtained using RNA sequencing and methylation-sensitive nanopore long-read sequencing. For example, we assembled 1.28 Mbp of Y-chromosome sequence, the most of any marsupial, and in doing so revealed at least one Y-linked gene not reported in other mammals to date. Differential expression patterns of these Y-linked genes compared to their X-linked counterparts provide considerable support for the hypothesis that they have undergone adaptive evolution on the Y-chromosome.

Our assembly also revealed the genetic sequence of vulpeculin, a major urinary protein, and its previously unknown homologue, ganderin. Possums are thought to use chemical signals as their primary form of social communication[70], with males, in particular, engaging in territory-marking behaviour, where urine is deposited along the length of a log or branch. Male-biased expression of *vulpeculin* and *ganderin* therefore supports the hypothesis that these proteins carry and prolong pheromone signals in possums and other marsupials. Restriction of expression to post-weaning also makes sense, given both the metabolic cost of producing lipocalin molecules for urinary excretion[71] and the fact that possum pouch young interact only with their mother, so do not need to produce urinary chemical communication signals.

We used RNA-sequencing data as a cost-effective reduced representation means to sample population dynamics in the Dunedin region, as well as assemble mitochondrial sequences. Together, this analysis showed considerable admixture has occurred since possums were introduced to New Zealand, and our work adds to the New Zealand sites known for interbreeding of Tasmanian and mainland sub-

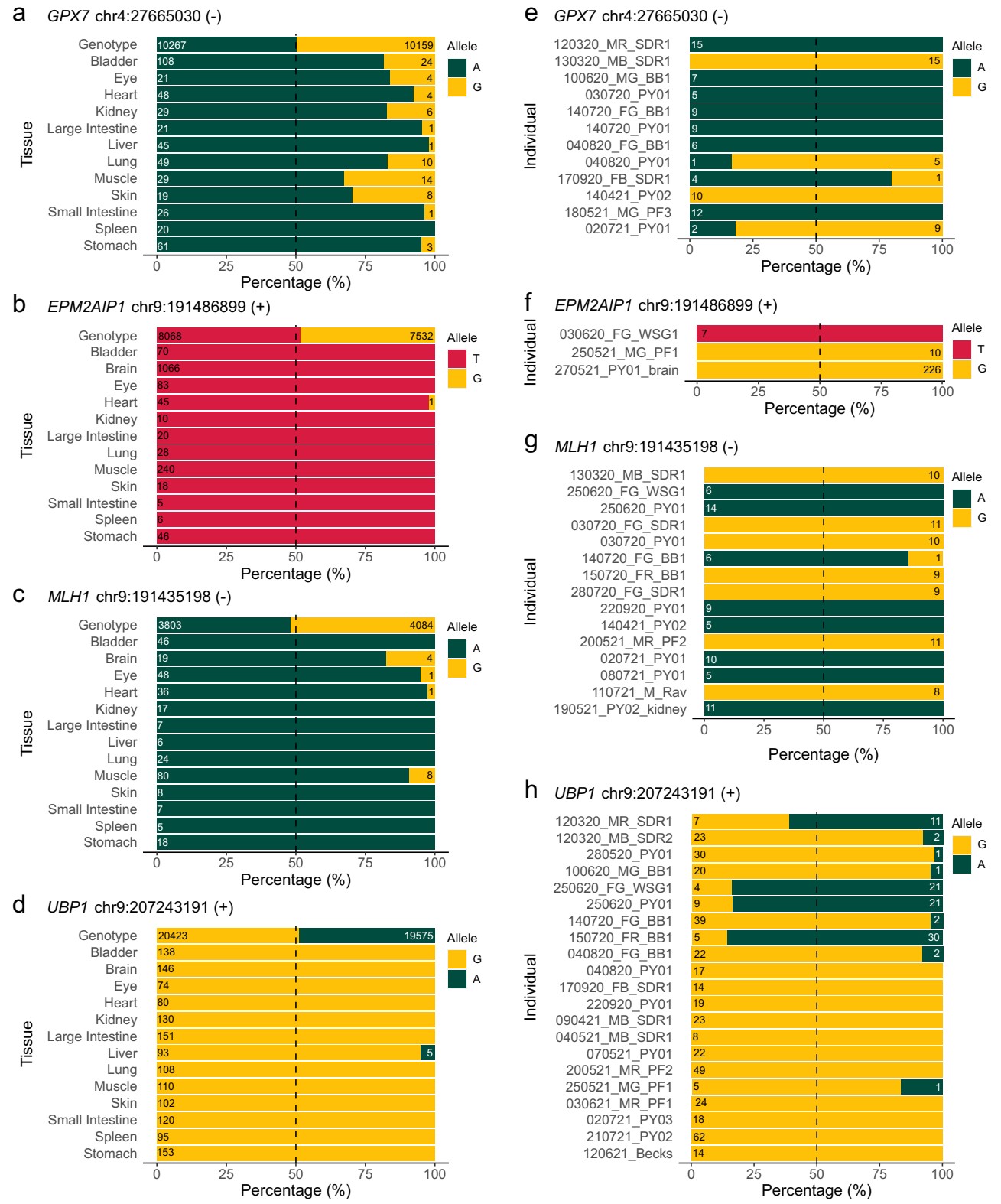

**Fig. 5 | Novel allele-specific methylation sites correlate with monoallelic expression of neighbouring genes.** Monoallelic expression of single-nucleotide polymorphisms within *GPX7* (**a**, **e**), *EPM2AIP1* (**b**, **f**), *MLH1* (**c**, **g**) and *UBP1* (**d**, **h**) in RNA-sequencing data from Sandy's tissues (**a**–**d**) and heterozygous individuals (**e**–**h**). Values represent read depth (≥5). Source data are provided as a Source Data file.

types[45]. We could trace possums in our southern New Zealand dataset to closely related modern Australian populations from Tasmania and south-east mainland Australia. This is in line with historical records— the Otago Acclimatisation Society documented the importation of breeding possums from at least Leongatha in Victoria, and Hobart in Tasmania (both directly from Hobart and via the Southland population also established from Hobart), with additional introductions also possible[22,72,73].

DNA methylation analysis further enhanced the utility of the possum genome sequence. Precise quantitation of methylation in

**Table 1 | Summary of parent-specific monoallelic expression data, with potential gene function, indicated**

| Gene | Heterozygous individuals displaying monoallelic expression | Parent-specific expression? | Number of duos identified | Function—human and mouse studies |
|------|------|------|------|------|
| GPX7 | 19 | Yes—Maternal | 7 | Protects from oxidative stress, suppresses growth[77] |
| EPM2AIP1 | 7 | Yes—Paternal | 5 | Involved in glycogen storage, promotes growth[79] |
| MLH1 | 31 | Yes—Paternal | 5 | DNA mismatch repair protein, stable epiallele[82] |
| UBP1 | 35 | Yes—Paternal | 4 | Embryonic lethal promotes placental growth[78] |

PGCs using bisulfite sequencing not only confirms that global germline epigenetic erasure occurs in marsupials[60,61,69], but also validates patterns of imprinting at ASM sites that we identified by phased nanopore sequencing of somatic tissue. In addition, the discovery of cells with erased imprints provides undeniable molecular proof that bona fide PGCs can be isolated from possum during post-natal life. This is significant because manipulation of PGCs may prove to be the most efficient way to genetically modify the marsupial germline, as is the case in avians[74].

Using long-read methylation sequencing as a starting point, we discovered 4 genes with parent-specific imprinted expression across a wide range of tissues, albeit with two genes (UBP1 and GPX7) showing leaky monoallelic expression, as previously reported for some marsupial imprinted genes[75]. Associated with these imprinted genes was parent-specific methylation—two instances where the maternal chromosome was methylated (like most imprinted loci), and one example of paternal methylation (GPX7). Interestingly, GPX7 ASM is located in an intron, similar to other known paternally methylated imprint control regions, and unlike maternal imprint control regions which are generally located at transcription start sites[76]. According to the parental-conflict hypothesis, maternal-specific expression of GPX7 in possum implies it should function as a growth suppressor. Indeed, mice deficient in GPX7 exhibit markedly greater fat mass and adipocyte hypertrophy[77]. In contrast, paternally expressed genes, like UBP1 and EPM2AIP1 should normally enhance growth. Inline with this, knockout of UBP1 in mice drives intrauterine growth retardation at embryonic day 10.5, with a significant reduction in the thickness of the labyrinthine layer of the placenta[78]. Likewise, mice defective in EPM2AIP1 are smaller than wild-type littermates and resist weight gain in adulthood[79].

The MLH1 gene does not have an obvious link to the parental-conflict hypothesis and therefore may be a 'bystander' caught in the imprinted regulation of closely linked EPM2AIP1. Nevertheless, the discovery that MLH1 is imprinted in at least one divergent mammalian taxa is significant. In humans, MLH1 is well-known for its role in DNA mismatch repair and is frequently mutated in Lynch Syndrome, an autosomal dominant cancer predisposition condition[80]. A proportion of Lynch Syndrome cases do not have genetic changes associated with them, rather, appear to be inactivated through repressive DNA methylation in the promoter[81,82]. One feature that makes human MLH1 epimutation curious is the remarkable stability by which MLH1 methylation is retained once laid down, affecting a patient for life as well as having the potential to be inherited[82]. The number of loci possessing true epigenetic bistability in the form of DNA methylation is rarer than often assumed, affecting a small number of germline-expressed genes, as well as imprinted genes[83]. We speculate that the epigenetic bistability of MLH1 in human cancer could be related, at least to some extent, to an evolutionary history featuring genomic imprinting.

Marsupials are often thought of as a 'lesser mammal', because they display many ancestral and intermediate traits shared with reptiles and birds. Here we show that marsupials possess at least 4 imprinted genes, spread over 3 separate loci, that are not known to have imprinted expression in eutherian mammals. It is likely more imprinted genes in possum will be uncovered—for example, just as H19 was not detected by our pipeline, it is possible other ASM sites and

imprinted genes will have been missed due to a lack of heterozygosity (Fig. S5a). In addition, we found 9 regions with ASM that have evidence for monoallelic expression in at least the brain, muscle, or liver in Sandy; but require further testing to be sure of their imprint status. Interestingly, of these ASM sites, 4 are located near genes recently identified to have imprinted expression in opossum[17] (Fig. 6d, bottom panel; Supplementary Data 3). Given significant numbers of novel imprinted genes in both possum and M. domestica, it may well be that from the perspective of genomic imprinting, rather than being a 'lesser mammal', marsupials possess their own unique suite of imprinted genes tailored to their specific developmental and reproductive characteristics.

In summary, the possum genome assembly provides insights into mammalian chromosome and genome evolution. When coupled with population and functional genomics in the form of RNA-sequencing and methylation analysis, significant insight into the invasion history of possums in New Zealand has been gained, but also new angles from which to control and eradicate their population, including PGC characterisation and chemosensory genes such as vulpeculin and ganderin. Equally, we have further consolidated the possum as an important model for marsupial development and reproduction, describing the metabolic gene changes associated with extended lactation and weaning using transcriptomics, as well as uncovering novel mammalian imprinted genes.

## Methods

### Sample collection and metadata
Tissue samples were sourced from freshly deceased possums killed as part of a pest-control programme, and therefore not require animal ethics oversight according to guidelines issued by the NAEAC, Occasional Paper No 2, 2009 (ISBN 978-0-478-33858-4). Additional samples were collected from unused tissue belonging to captive possums euthanised from another study. Capture, husbandry and manipulation in that study were approved by the Otago Animal Ethics Committee (AUP-19-75, AUP-20-10). Further samples for mitogenome analysis were sourced from a prior study[47].

Pouch young was developmentally staged using the nomogram of Lyne and Verhagen[2]. Pouch young >120 days of development were challenging to the stage, and for this reason, have been referred to as juveniles. Upon collection, tissue was added to RNAlater (Invitrogen, AM7020) and placed on ice until storage at +4 °C for 24 h, followed by long-term storage at −80 °C. Alternatively, tissue was processed in order to collect germ cells (see below), or samples for DNA extractions (from ears, tails or limbs) were snap-frozen in liquid nitrogen and stored at −80 °C (see below). Sperm were isolated from the epididymis of an adult possum by allowing sperm to diffuse out into warm 1x PBS (37 °C) for 1 h. Plasma was removed by centrifugation at 240 × g for 2 min and washing with 1x PBS, repeated twice. The resulting pellet was resuspended in 1x PBS, passed through a 20 μm cell strainer and stored at −80 °C. Metadata for each sample used in this work is given in Supplementary Data 5.

### DNA analyses
**DNA extraction and library preparation for de novo genome assembly.** Ultra-high molecular weight DNA (uHMW) was extracted using the agarose plug Bionano Genomics protocol from Sandy's

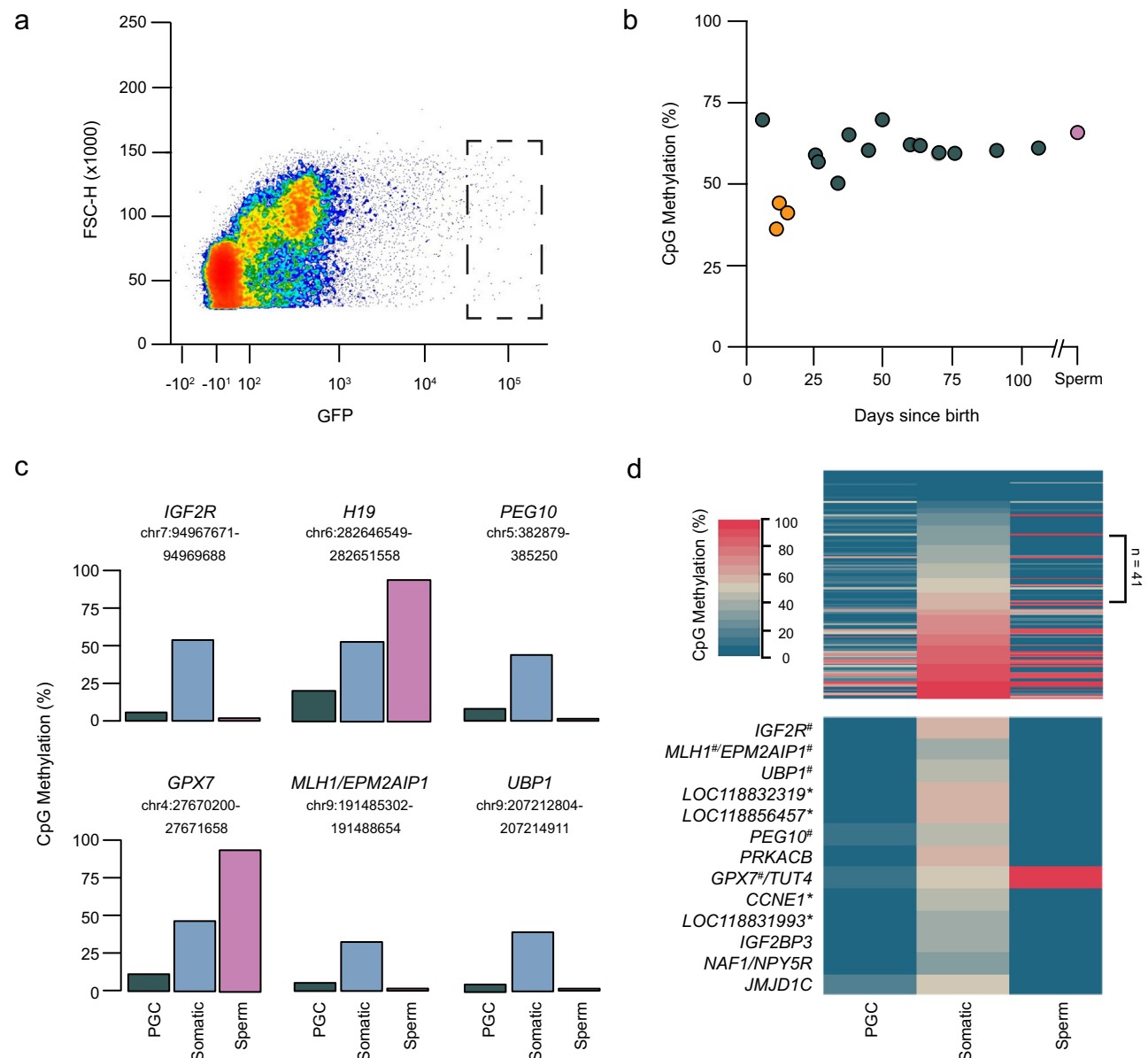

**Fig. 6 | Male germline reprogramming of imprinting and the epigenome. a** Flow cytometry plot of SSEA1 (GFP) labelled primordial germ cells (PGCs) purified from a representative individual (13 dpp). Dots are individual cellular events and colours represent data density (red=highest). Cellular events are plotted against forward scatter height (FSC-H) and intensity of GFP fluorescence. Dashed box represents the gating strategy used to isolate GFP-positive PGCs. GFP-negative (somatic) cells were sorted from outside of the gated range (from cluster on left-hand side). **b** Genome-wide CpG methylation (%) of male germ cells (GFP-positive) from pouch young gonads between 7 and 106 days since birth (*n* = 16; dark green and orange dots) and from adult sperm (*n* = 1; pink dot). Orange samples represent individuals with reduced global DNA methylation ('hypomethylated'). Biologically independent samples (*n* = 15; orange dots and subsequent dark green dots) from

forearm muscle. *n* = 3 sequencing runs were used for statistical analysis. **c** CpG methylation (%) at ASM sites for PGCs (GFP-positive, dark green) and somatic cells (GFP-negative; light blue) of two 'hypomethylated' individuals (13 and 16 dpp) and for adult sperm (pink). **d** Top panel: heatmap of methylation at all ASM sites (excluding those with missing data, *n* = 32) in PGCs, somatic cells and sperm. ASMs with somatic methylation 30–57% are indicated (*n* = 41). Bottom panel: heatmap of methylation at ASM sites in PGCs, somatic cells and sperm associated with imprinted genes confirmed by our work and previously[53,54] (#), ASMs near genes recently identified to have imprinted expression in opossum[17] (*) or candidate imprinted genes displaying monoallelic expression in Sandy's brain, liver or muscle. Source data are provided as a Source Data file.

forearm muscle. uHMW DNA quality was assessed by a Pulsed Field Gel assay and quantified with the Qubit DNA High Sensitivity dsDNA Kit and Qubit Fluorometer (Invitrogen, Q32854 and Q33238).

**PacBio continuous long reads (CLR).** Five micrograms of uHMW DNA were sheared using a 26G blunt end needle (PacBio protocol, 101-181-000, Version 05). A large-insert PacBio library was prepared using the Pacific Biosciences Express Template Prep Kit v1.0 (101-357-000) following the manufacturers' protocol. The library was then size-selected

(>15 kb) using the Sage Science BluePippin Size-Selection System. The final library was sequenced on PacBio Sequel smrtcells (~60x coverage).

**10x Genomics linked reads.** The same unfragmented uHMW DNA was used to generate a linked-reads library with the 10x Genomics Chromium (Genome Library Kit & Gel Bead Kit v2, 120258; Genome Chip Kit v2, 120257; i7 Multiplex Kit, 120262). This library was then sequenced on an Illumina Novaseq S4 (150 bp paired end reads).

**Bionano genomics optical maps.** Unfragmented HMW DNA was also used for Bionano Genomics optical mapping. Briefly, DNA was labelled using the Bionano Prep Direct Label and Stain (DLS) Protocol (30206E) and run on one Saphyr instrument chip flow cell. 1.1 Gb of data was generated and optical maps were assembled using Bionano Access.

**Arima Genomics Hi-C proximally ligated DNA.** Hi-C libraries were generated by Arima Genomics (https://arimagenomics.com/) using the Arima-HiC kit v1 (A510008) and sequenced on a HiSeq X (Illumina) at -147x coverage following the manufacturer's protocols.

**DNA extraction for all other analyses.** DNA was extracted from tissue (~3 mm tissue punches−liver, ear or tail) using a Bio-On-Magnetic-Beads (BOMB) protocol that utilises solid-phase reversible immobilisation (SPRI) carboxyl-coated Sera-Mag Magnetic SpeedBeads (Cytvia, 45152105050250)[84]. Tissue was lysed in 300 µL TNES buffer (100 mM Tris pH 8.0, 25 mM NaCl, 10 mM EDTA, 1% w/v SDS) with 5 µL Proteinase-K (20 mg/mL) at 55 °C overnight. Lysates were then mixed with 6 M Guanidinium Thiocyanate (GITC), SPRI beads suspended in 1x TE buffer (10 mM Tris pH 8.0, 1 mM EDTA), and isopropanol in a ratio of 1:2:2:4, respectively. For DNA extraction from sperm, the frozen pellet (described above) was resuspended in GITC supplemented with 10% v/v ß-mercaptoethanol as a reducing agent[85]. For all tissues, after a 5 min incubation to allow DNA to bind to the magnetic beads, tubes were placed on a neodymium magnetic rack for ~5 min until the solution was clarified. The supernatant was removed and the DNA bound to the magnetic beads was washed once with isopropanol and twice with 70% ethanol, and then left to air dry on the magnetic rack. DNA was eluted from the magnetic beads in nuclease-free water or 1x TE buffer and concentrations were measured using the Qubit DNA High Sensitivity dsDNA Kit and Qubit Fluorometer.

**PCR amplification.** Amplification of DNA for Y-chromosome analyses, mitochondrial DNA library preparations and amplicon sequencing for imprinted SNP analysis (see below) was performed using Phusion High-Fidelity DNA Polymerase (New England Biolabs, M0530L) according to the manufacturers' protocol. Approximately 10−25 ng of DNA was used in the following PCR mix: 1x HF Buffer with 1.5 mM MgCl₂, 0.2 mM of each dNTP, 0.5 µM of each primer, and 0.02 U/µL units of Phusion. Primer sequences and PCR cycling conditions used for each amplification are specified in Supplementary Table 3.

**Preparation and sequencing of mitochondrial DNA libraries.** The mitochondrial genomes not assembled from RNA-sequencing data were amplified using five overlapping primer pairs (Supplementary Table 3). For each sample, the five PCR products were pooled in equivolume amounts and cleaned up and size-selected using 0.8X ratio of SPRI beads diluted in standard PEG buffer (18% w/v polyethylene glycol 8000 (PEG), 1 M NaCl, 10 mM Tris (pH 8.0), 1 mM EDTA, 0.05% v/v Tween-20[84]). The concentration of each resulting PCR pool was determined using the Qubit DNA High Sensitivity dsDNA Kit and Qubit Fluorometer. The NEBNext Ultra II FS DNA Library Prep Kit for Illumina (New England Biolabs, E7805L) was used with 50 ng of each PCR pool to make a DNA library with the following modifications: the reaction volumes were reduced to a ¼ of the standard reaction volume, a fragmentation/end prep time of 12 min, 1.5 µM Y-adapter (see below) was used for adapter ligation. The resulting libraries were amplified with 0.4 µM indexed Truseq-type oligos for Illumina sequencing with the following conditions: 98 °C for 30 s, followed by 5 cycles of 98 °C for 10 s, 65 °C for 75 s, and a final elongation of 65 °C for 5 min. The unique barcode for each sample is given in Supplementary Data 5. The final libraries were pooled in equimolar amounts and sequenced on the iSeq100 (Illumina) to generate 150 bp paired-end reads.

The Y-adapter includes the PE_Yadapter_1 (/5Phos/GAT CGG AAG AGC ACA CGT CTG AAC TCC AGT C) and PE_Yadapter_2 (ACA CTC TTT CCC TAC ACG ACG CTC TTC CGA TC*T) which were pooled 1:1 in a 100 µL volume to make a 50 µM stock. Added to this pool was 1 µL of 5 M NaCl (50 mM final) for adapter annealing with the following conditions: 95 °C for 2 min, followed by ramping 95 °C to 25 °C over 45 min, at 0.1°/s (or 6°/min). The annealed Y-adapter was then diluted to 1.5 µM.

**Preparation and sequencing of nanopore libraries.** Following DNA purification from Sandy's forearm tissue (BOMB extraction protocol, above), nuclease-free water was added to a final volume of 400 µL in order to reduce contaminants. The sample was treated with 2 µL of 10 mg/mL RNaseA (to remove any remaining RNA) and placed at 37 °C for 5 min. To remove the RNaseA and any remaining proteins, the sample was treated with 2 µL of 20 mg/mL Proteinase-K at 50 °C for 5 min. Chloroform:isoamyl alcohol (IAA) (24:1) was then added (400 µL) to the DNA sample, which was mixed by gentle rotation at room temperature for 5 min. The sample was then spun at 15,320 × g for 5 min and the top phase (~350 µL) was transferred to a new tube containing 350 µL Chloroform:IAA (24:1). The sample was mixed again by gentle rotation at room temperature for 5 min, followed by centrifugation at 15,320 × g for 5 min. The top phase (~300 µL) was transferred to a new tube and 0.1X volume (30 µL) of 3 M NaOAc pH 5.2 followed by 330 µL of 100% ice-cold ethanol was added and the tube was gently inverted. The sample was then spun at 15,320 × g for 5 min and the supernatant was removed. The resulting pellet was washed with 1 mL ice-cold 70% ethanol and spun at 5200 × g for 5 min. The ethanol wash was removed and the pellet was air-dried before resuspending the DNA in 100 µL 1x TE buffer. Sandy's DNA was sequenced on five R9.4.1 MinION flow cells (Oxford Nanopore Technologies) prepared with the Rapid Sequencing Kit (SQK-RAD004; run 1, with 400 ng of DNA) or the Ligation Sequencing Kit (SQK-LSK110; run 2-5, with 1 µg of DNA). For run 2-5 with DNA prepared with the Ligation Sequencing Kit, ~15 fmol of DNA was loaded onto the MinION flow cell.

**Amplicon sequencing for imprinted SNP analysis.** A dual-indexing, four-primer PCR-based assay was used for amplicon sequencing[86]. Primers contain a 'handle' sequence such that during amplification the handle is incorporated into the amplicon (Supplementary Table 3). Following the first round of PCR amplification (as described above), ~¼ of the PCR reaction was cleaned up and size-selected using a 0.9x SPRI beads diluted in standard PEG buffer[84]. The DNA was eluted in 10 µL nuclease-free water and 2−5 µL was used as a template in a second round of PCR amplification (as described above) with 0.2 µM of each second step primer (indexed Truseq-type oligos for Illumina sequencing; Supplementary Table 3). PCR cycling parameters for the second step PCR were: 98 °C for 3 min, 5 cycles of 98 °C for 10 s, 62 °C for 20 s, and 72 °C for 15 s. A final elongation step was performed for 2 min at 72 °C. The final libraries were pooled in equimolar amounts, cleaned-up and size-selected using a 0.9x SPRI beads diluted in standard PEG buffer and sequenced on the iSeq100 (Illumina) to generate 150 bp paired-end reads.

**Post bisulfite adapter tagging (PBAT) of sperm DNA.** Bisulfite-converted genomic libraries with sperm DNA were prepared using a modified PBAT method[87]. Approximately 50 ng of DNA was bisulfite-converted using the EZ Methylation-Direct MagPrep kit (Zymo, D5044), but reducing the reaction volume to one ¼ of that stated in the protocol (to allow for all reaction and clean-up steps to be performed in PCR tubes). Bisulfite-converted DNA was eluted from the magnetic beads in 20 µL elution buffer and transferred to a new tube for first-strand synthesis with a 5′-biotinylated adapter primer bearing seven random nucleotides at its 3′ ends (BioP5N7, biotin-ACACTCTTTCCCTACACGACGCTCTTCCGATCTNNNNNNN). The first

strand synthesis product was then purified using Dynabeads™ M-280 Streptavidin beads (Invitrogen, 11205D) and alkaline denaturation. Second-strand DNA synthesis involved the immobilised first-strand DNA and a second adapter primer also bearing seven random nucleotides at its 3′ ends (P7N7, GTGACTGGAGTTCA-GACGTGTGCTCTTCCGATCTNNNNNNN). The dsDNA product was then PCR amplified (13 cycles) using 1X HiFi HotStart Uracil + Mix (Roche, KK2802) with 0.4 μM primer (indexed Truseq-type oligos for Illumina sequencing), so that sample-specific barcodes and sequences required for Illumina flow-cell binding were added to the library. The unique barcode for this sample is given in Supplementary Data 5. The integrity of the library was assessed by agarose gel electrophoresis before size-selecting using standard PEG-diluted SPRI beads (0.8X). The pooled library was sequenced on the NextSeq2000 (Illumina) at the Otago Genomics Facility to generate 50 bp single-end reads.

**Gonad collection and germ cell labelling.** Gonads were collected from male pouch young (n = 17; Supplementary Data 5) and germ cells were isolated from the gonads as previously described[88]. Briefly, gonads were torn using needles in a droplet of TrypLExpress (Gibco, 12605010) and incubated at 37 °C for 20 min. Tissue clumps were triturated manually by gentle strokes with a P1000 pipette tip and then with a P200 pipette tip. TrypLExpress was diluted with 5% FBS (in PBS), filtered through a 40 μm cell strainer, and washed twice in PBS (centrifugation 240 × g for 2 min). The resulting pellet was resuspended in Zombie Near-Infra Red Viability Dye (BioLegend, 423105) and incubated at room temperature for 20 min. The lysate was washed in 5% FBS (in PBS), then resuspended in SSEA1 monoclonal antibody (Invitrogen, MC-480) conjugated to DyLight 488 (1:300 in 5% FBS (in PBS); Invitrogen, MA1-022-D488) on ice for 30 min. The lysate was then washed, resuspended in sort buffer (5% FBS, 0.5 mM EDTA in PBS), and kept on ice prior to cell sorting. Validation of the SSEA1 monoclonal antibody for germ cell specificity in brushtail possum was previously undertaken[74].

**Germ cell isolation using fluorescence-assisted cell sorting (FACS).** Germ cells were isolated from the cell lysate using a BD FACSAria Fusion which detects GFP using a 488 argon laser (BD Biosciences). A gonadal cell profile was produced using forward scatter (FSC) and side scatter (SSC) and was gated to exclude doublets and dead cells. Germ cells were identified as a small sub-population that was very bright (GFP > $10^4$) with higher SSC (Fig. 6a and Supplementary Data 6; approx. 0.1% of total cells). This population was gated conservatively, sorted into 0.2% Triton-X and stored at −80 °C. The number of GFP-positive sorted cells varied between 3 and 516 per sample (Supplementary Data 5). GFP-negative cells (somatic cells) were also sorted from each individual as negative controls.

**Low cell input bisulfite sequencing.** Bisulfite conversion and library preparation followed the single-cell bisulfite sequencing protocol[89], substituting modified PEG-diluted SPRI beads for AMPure XP beads. This protocol is similar to PBAT (above) but carries SPRI beads through most steps with the addition of modified PEG buffer (18% w/v PEG 8000, 2.5 M NaCl, 10 mM Tris (pH 8.0), 1 mM EDTA, 0.05% v/v Tween-20) for size-selection steps. GFP-positive germ cells and GFP-negative somatic cells from each individual were incubated with 0.5 μL of 20 mg/mL Proteinase-K on ice for 15 min followed by the addition of modified PEG-diluted SPRI beads for a further incubation of 20 min at room temperature. Tubes were placed on a neodymium magnetic rack for ~5 min until the solution was clarified, the supernatant removed and beads washed twice in 90% ethanol. The DNA was eluted into 10 μL of 10 mM Tris (pH 8.0), but kept with the beads to avoid loss of the DNA. Bisulfite conversion was then performed as for PBAT (see above), carefully adding the conversion reagent to the DNA/bead suspension to prevent disturbing the beads. The resulting bisulfite-converted DNA

underwent five rounds of first-strand synthesis using the 5′-biotiny-lated adapter primer (BioP5N7) with intermediate heat denaturation of the DNA fragments[89]. Exonuclease I (New England Biolabs, M0293S) was added to the first strand synthesis product to remove any remaining adapter primer, and the product was cleaned up and size selected with modified PEG-diluted SPRI beads (0.8X). Second-strand synthesis was performed in the presence of SPRI beads and was otherwise the same as for PBAT. The second strand product (50 μL) was stimulated to bind the SPRI beads by the addition of 50 μL of 10 mM Tris-HCl and 80 μL of modified PEG buffer (0.8x), followed by on-bead PCR amplification as for PBAT (14 cycles for 100 cells, 16 cycles for <100 cells). The unique barcode for each sample is given in Supplementary Data 5. PCR products were size selected by addition of modified PEG buffer (0.8x) and integrity of each library was assessed by agarose gel electrophoresis. Libraries were then pooled in equimolar amounts and sequenced on the iSeq100 to generate 150 bp paired-end reads. Samples of GFP-positive cells (and their GFP-negative somatic cells) (n = 2; 13 and 16 dpp) were further sequenced (high coverage) on the NextSeq2000 as for the sperm sample (above).

## RNA analyses

**RNA extraction and library preparation for de novo assembly.** In order to assist with annotation of the genome, long-read RNA-sequencing was performed on muscle, brain and testis tissue using the PacBio IsoSeq method as per VGP pipelines[27].

**RNA extraction and preparation for all other analyses.** Total RNA was extracted from tissue samples (~50 mg) that were stored in RNA-*later* using TRIzol Reagent (Invitrogen, 15596026) following the manufacturer's guidelines. RNA samples were treated to remove any contaminating DNA using the Turbo DNA-*free* kit (Invitrogen, AM1907) according to the protocol. The concentration of DNase-treated RNA was determined using the Qubit RNA High Sensitivity Kit (Invitrogen, Q32852) with the Qubit Fluorometer.

**Library preparation of short-read RNA-sequencing libraries.** Poly(A)-enriched, strand-specific RNA-sequencing libraries were prepared using either the NEXTFLEX Rapid Directional RNA-Seq Kit 2.0 and NEXTFLEX Poly(A) Beads (2.0) (Perkin Elmer, NOVA-5198 and NOVA-512991) or the NEBNext Ultra II Directional RNA Library Prep Kit for Illumina and NEBNext Poly(A) mRNA Magnetic Isolation Module (New England Biolabs, E7760L and E7490L) according to the manufacturers' guidelines with the following modifications. For the NEXTFLEX kit, 400 ng of DNase-treated RNA was used as input, fragmentation time was 12 min, 0.48 μM of NEXTFLEX RNA-Seq 2.0 Unique Dual Index Barcodes (Perkin Elmer, NOVA-512920) was used for adapter ligation and 12 cycles for PCR amplification. The Unique Dual Index for each sample is given in Supplementary Data 5. The final libraries were pooled in equimolar amounts and run on 5x lanes of the HiSeq 2000 (Illumina) at the Otago Genomics Facility to generate 100 bp single-end reads. For the NEB kit, the reaction volumes were reduced to a ¼ of the standard reaction volume with 250 ng of DNase-treated RNA used as input, fragmentation time was 12 min, 3.125 μM of NEXTFLEX RNA-Seq 2.0 Unique Dual Index Barcodes (Perkin Elmer) was used for adapter ligation and the following modified PCR protocol because a non-NEB adapter was used: 98 °C for 30 s, followed 11 cycles of 98 °C for 10 s, 60 °C for 30 s, 65 °C for 45 s, and a final elongation of 65 °C for 5 min. The Unique Dual Index for each sample is given in Supplementary Data 5. The final libraries were pooled in equimolar amounts and run on the NextSeq2000 at the Otago Genomics Facility to generate 50 bp single-end reads.

**cDNA synthesis and reverse transcription-PCR (RT-PCR) amplification.** For cDNA synthesis, 500 ng of DNase-treated RNA was used with the RevertAid First Strand cDNA Synthesis Kit (Thermo Scientific,

K1632) according to the manufacturers' guidelines for random hexamer primed synthesis and GC-rich RNA templates. For all RNA samples processed, a minus reverse transcriptase (-RT) control was included to determine if any resulting amplification could be from genomic DNA. No amplification was seen in any of the -RT control samples tested.

Amplification of cDNA for RT-PCR analyses was performed using Phusion High-Fidelity DNA Polymerase (New England Biolabs) according to the manufacturers' protocol. Each cDNA sample was diluted 2-fold with nuclease-free water and 2 μL was used in a 20 μL volume in the following PCR mix: 1X HF Buffer with 1.5 mM MgCl₂, 0.2 mM of each dNTP, 0.5 μM of each primer, and 0.02 U/μL units of Phusion. Primer sequences and PCR cycling conditions used for each amplification are specified in Supplementary Table 3. In order to sequence the resulting PCR products and determine allele-specific expression, the amplicons were cleaned-up size-selected, and processed as for amplicon sequencing for imprinted SNP analysis (see above).

### Bioinformatic analyses

Reference samples (RNA-sequencing and whole-genome sequencing (WGS)) were included in various bioinformatic analyses and are outlined in Supplementary Data 5. These included samples from the Australian locations Karta Pintingga (Kangaroo Island; KAN), lutruwita (Tasmanian; TAS), and New South Wales (NSW)[50], as well as Aotearoa New Zealand (NZ)[49].

**Whole-genome sequencing analysis.** Paired FASTQ files for Sandy's WGS data were obtained from GenomeArk (https://genomeark.s3. amazonaws.com/index.html?prefix=species/Trichosurus_vulpecula/ mTriVul1/genomic_data/10x/). For the TAS WGS sample, an additional initial step removing 10X adaptors was performed using an *in house* script. Paired reads were trimmed using Trim Galore! (v0.6.7) in a two-step process: first, to remove adapters, and second, to remove low-quality base calls (Phred score <20) (https://github.com/FelixKrueger/ TrimGalore). The reads were mapped to the possum reference genome GCA_011100635.1 with the BWA-MEM algorithm (v0.7.17)[90]. The SAM files obtained were sorted and converted to BAM files using SAMtools (v1.15.1)[91]. To determine the sequencing depth of Sandy's WGS data, non-overlapping 1 kilobase windows were created for all chromosomes and contigs using the R package GenomicRanges (v1.38.0) and the number of reads mapping to each window was determined with the coverage command from BEDtools (v2.30.0)[92].

**RNA-sequencing expression and coverage analysis.** Raw reads were trimmed using Trim Galore! (v0.6.7) to remove adapters and low-quality base calls (Phred score <20) (https://github.com/FelixKrueger/ TrimGalore). Trimmed reads were mapped to the possum reference genome GCA_011100635.1 using HISAT2 (v2.2.1)[93].

For expression analysis, the HISAT2 generated BAM files were imported into SeqMonk (v1.48.0) (https://www.bioinformatics. babraham.ac.uk/projects/seqmonk/) for visualisation. The SeqMonk RNA-seq quantitation pipeline was employed with uniquely mapped reads to generate a set of probes covering every gene in the genome, which were quantitated based on the number of reads falling within this annotation. The resulting log-transformed counts were corrected for the total number of sequences in each dataset to generate an RPKM (log₂) value for each gene. Two samples (020721_PY01 and 210521_PY01_liver) were removed from further expression analyses due to poor mapping efficiencies and RNA QC plot analysis assessed via SeqMonk (v1.48.0)).

**SNP calling using short reads.** Short-read SNP calling of RNA-sequencing and WGS-mapped data (see mapping details above) was performed following the GATK's Best Practices (v4.2.6.1)[94]. Duplicate reads (from the BWA-MEM mapping) were marked using the Mark-Duplicates GATK module and read group tags were added using the addreplacerg function from SAMtools (v1.15.1)[91]. The SNPs were called into gVCF files for each sample using the HaplotypeCaller GATK module. Individual gVCFs were combined using the CombineGVCFs GATK module and genotypes of the samples were obtained with the GenotypeGVCFs GATK module. High-quality SNPs were filtered with VCFtools (v0.1.15) using the following parameters: --minQ 30 --minmeanDP 10 --max-missing 0.95[95]. After running this pipeline, a genome-wide dataset comprising 92 samples and 2310 SNPs was obtained.

**Mitochondrial genome assembly.** Reads from mitochondrial DNA libraries were trimmed using Trim Galore! (v0.6.7) to remove adapters, low-quality base calls and short reads (Phred score <20, length <35) (https://github.com/FelixKrueger/TrimGalore). Trimmed reads from mitochondrial DNA libraries and RNA-sequencing libraries (see above) were then mapped to the possum reference mitochondrial genome (NC_003039.1) using BWA-MEM algorithm (v0.7.17)[90]. The SAM files obtained were sorted and converted to BAM files using SAMtools (v1.15.1)[91]. For RNA-sequencing derived assemblies a FASTQ consensus sequence was generated from each of these files using the mpileup command from BCFtools (v1.15.1) and seqtk (v1.3) was used to mask (replace with 'N') all base calls with Phred score values less than 20. Consensus sequences of assembled mitochondrial genomes are given in Supplementary Data 7.

**Mitochondrial genome phylogenetic and spatial analyses.** Assembled mitogenomes derived from RNA-sequencing and mitochondrial DNA libraries were aligned together with KAN, TAS and NSW reference samples using the default Geneious v2022.2.2 (https://www.geneious. com) algorithm. Because samples derived from RNA-sequencing had significantly higher levels of missing data for the control region/origin of the light strand, all tRNAs, and both rRNAs, phylogenetic analyses were restricted to protein-coding regions (although some genes had significantly more missing data in RNA-sequencing samples, the maximum missing data for any sample for any protein-coding gene was 1.24%). Protein-coding genes were scrutinised to ensure no premature stop codons/frame-shift mutations were present, indicative of inadvertent incorporation of nuclear-embedded mitochondrial DNA sequences (NUMTs), and maternally-related individuals were screened to ensure they had identical haplotypes[96].

Phylogenetic trees were built utilising the codon-partitioned mitochondrial DNA protein-coding alignments. Maximum likelihood (ML) trees were constructed via RAxML v 8.2.12[97] with 100 bootstrap replicates[98]. Two separate runs were used to confirm convergence, utilising the GTRCAT model of substitution. After confirming topological convergence based on the bipartitions file, we utilised Run 1 for analyses as it had the highest likelihood. We verified our ML results using BEAST v 2.6.6[99], again implementing two runs to check the convergence of 250,000,000 states, sampling every 10,000 states. We implemented a gamma site model (4 categories, initial shape 1.0, estimated), and TN93 with Kappa1 and Kappa2 estimated (initially set at 2.0). We also estimated site frequencies and a strict clock as we did not expect marked rate variation within species. We used a Coalescent Bayesian Skyline[100] as our tree model due to uncertainty about the population demography and linked the tree model across all partitions while allowing site and clock to vary. Following the removal of the initial 10% of states as a burn-in, we checked for convergence of parameters in Tracer v 1.7.2[101] and of topologies of the consensus trees constructed with TreeAnnotator (maximum clade credibility tree, common ancestor heights). After confirming convergence, the two runs were combined, checked for adequate ESS values, and a combined consensus tree was created.

The reference samples were used to define the origin of mitochondrial haplotypes within our Dunedin samples as either NSW, TAS, KAN, vs. 'other well-supported clades' not represented in our Australian full-length reference samples (note, although these latter clades did not match full-length mitochondrial reference sequence we had, they could be identified as being most similar to D-loop sequence originating from south-east mainland Australia[48]). Only one discrepancy between runs/methods was detected: Peninsula2018_022 fell within the NSW clade in both RAxML runs, but differed between the BEAST runs (NSW and TAS), with the combined BEAST run finding this sample falling within the TAS clade. However, the split between NSW and TAS clades was not strongly supported in our analyses (<61% bootstrap support). Given the overall concordance between methods, we based downstream analyses (i.e. haplotyping) on the RAxML Run 1 tree. We used R v 4.2.1 and the tidyverse v 1.1.2, ggmap v 3.0.0, and ggrepel v 0.9.1 packages to examine the spatial distribution of mitochondrial haplotypes in the greater Dunedin region[98].

**Nuclear population genomics.** Standardised multilocus heterozygosity (sMLH) was calculated for each of the samples using variant calls derived from the RNA-sequencing samples with R utilising the inbreedR, vcfR, reshape, and future.apply packages. Comparisons of sMLH between possums with different mitochondrial haplotypes were carried out in R[98] with the tidyverse and rstatix v 0.7.0 packages.

To calculate levels of admixture among Dunedin possums we used the high-quality SNP data obtained from the RNA-sequencing and WGS datasets. First, we removed closely related individuals using three consecutive rounds of sample filtering, setting a threshold of PI_HAT scores >0.25 obtained by PLINK v1.90b6.21[102]. This filtering was conducted separately for the NSW and Dunedin datasets. After sample removal, we obtained a dataset comprising 60 individuals and 2248 autosomal SNPs. Principal component analysis (PCA) was run on the filtered dataset using the pca command in PLINK. The number of genetic clusters best fitting the data was calculated by performing a 10-fold cross-validation in ADMIXTURE v1.3.0 (from $k = 2$ through $k = 10$), resulting in the highest support for two genetic clusters ($k = 2$)[103]. The results were drawn using the ggplot v3.3.6 package in the R language.

**Identification of allele-specific methylation by nanopore sequencing data.** The raw FAST5 files obtained for each sequencing run were base-called using Guppy v6.2.1 with the following parameters: guppy_basecaller -i FAST5 -c dna_r9.4.1_450bps_sup.cfg -x "cuda:0" -s FASTQ. Sequencing reads were trimmed to remove adapters and low-quality reads using Porechop (v0.2.4) and Filtlong (v0.2.0), respectively (https://github.com/rrwick/Porechop; https://github.com/rrwick/Filtlong). Reads with internal adapters, average Phred score <9 or length <500 were discarded. Trimmed reads were mapped to the possum genome reference GCA_011100635.1 with Minimap2 (v2.24) using the "map-ont" parameter[104] resulting in 14.05X coverage (Supplementary Table 4). The bedcov tool from SAMtools (v1.15.1)[91] was used to calculate the proportion of the genome that was covered at ≥5x and 10x depth, which was estimated to be 98% and 80.1%, respectively. CpG methylation calls were extracted using the call-methylation module from Nanopolish (v0.13.3)[105]. To perform methylation phasing we first identified SNPs using Clair (v2.1.1)[106]. Clair parameters for SNP detection were set at a threshold of 0.2 and the 122HD34 was selected as the precomputed model. SNVoter (v.1.0) was used to improve SNP detection from low-coverage regions[54]. Subsequently, SNPs were phased using WhatsHap (v1.1) with default parameters and the --ignore-read-groups option[107]. CpG methylation calls, variant calls and phasing information were used to obtain phased CpG methylation and mock bis-sequencing BAM files using NanoMethPhase (v1.0)[54]. Mock bis-sequencing BAM files were visualised with IGV (v2.12.3) using the bisulfite colour alignment mode[108]. Differentially methylated CpG sites between haplotypes were identified using the

DMLtest function of the DSS package (v2.34.0). Finally, differentially methylated CpG sites, called here allele-specific methylation (ASM) sites, were filtered at a threshold of ≥20 CpG and annotated with the closest gene/mRNA (±20 kb) using SeqMonk (v1.48.1) (https://www.bioinformatics.babraham.ac.uk/projects/seqmonk/).

**Monoallelic analysis of known and candidate genes.** To investigate the monoallelic expression of known and novel imprinted genes, SNPs within these genes were identified following SNP analysis of the nanopore datasets (see above; Supplementary Table 5) and confirmed by PCR and amplicon sequencing (see above). The SAMtools (v1.9)[91] mpileup utility (max depth 0) was then used to provide a summary of the depth of mapped reads at the SNP for each RNA-sequencing library (Supplementary Data 4).

**Identification of genome-wide monoallelic SNP expression.** SNP calling to identify heterozygous sites in the reference genome was performed using a similar approach to RNA-sequencing SNP calling, with the additional step of using the high-quality SNPs obtained from RNA-sequencing data for variant recalibration and filtering. In total, we obtained 11,481,987 variants for Sandy's reference genome (0.35%). Next, we used the CollectAllelicCounts function from the GATK toolkit (v4.2.6.1) to quantify the number of reads with the reference and alternative alleles in the RNA-sequencing data[94]. Allelic count matrices were obtained for brain, liver and muscle. To identify monoallelic expressed SNPs, we compared the observed counts with the expected counts assuming biallelic expression (1:1). The thresholds were a minimum depth of 10x in RNA-sequencing data and a Fisher's exact $P$-value < 0.05 after Bonferroni correction for the observed/expected values. SNPs were annotated using SeqMonk (v1.48.0) (https://www.bioinformatics.babraham.ac.uk/projects/seqmonk/) and the most recent annotation reference (GCF_011100635.1). Using this approach we identified 2871, 2562 and 1826 potential monoallelic expressed SNPs in brain, liver and muscle, respectively. These SNPs were then overlapped with the ASM sites, using a +/−1 Mb cut-off, which resulted in 477, 407 and 347 monoallelic expressed SNPs in brain, liver and muscle, respectively (corresponding to 91, 61 and 62 candidate genes, respectively). IGV (v2.12.3) was then used to visualise the SNPs in RNA-sequencing libraries of Sandy's tissues and candidate genes were removed if biallelic expression of additional SNPs was observed[108]. This resulted in 13 genes (17 including known imprinted genes and genes characterised in this study), associated with 9 different ASM sites, containing SNPs displaying monoallelic expression that we classify as candidate imprinted genes (Supplementary Data 3).

**DNA methylation analysis.** For low-coverage bisulfite sequencing, raw reads were trimmed using Trim Galore! (v0.4.2; https://github.com/FelixKrueger/TrimGalore) by removing 10 bp from the 5' end of all reads and removing low-quality base calls (Phred score <20). Bismark (v0.19.0)[109] with the -pbat option was used to map reads to the possum genome reference GCA_011100635.1. From the Bismark output report, global methylation in the CG dinucleotide context was calculated as the proportion of total methylated cytosines over total cytosines. The bisulfite non-conversion rate was estimated by calculating the proportion of non-CG methylation; by this measure, all libraries must have had a bisulfite conversion efficiency of at least 96.02% (Supplementary Data 5). PGC libraries producing <400 CG calls or <35% mapping efficiency were excluded from the analysis. Based on CpG methylation levels, PGC samples were categorised into three groups: 'pre' methylation loss (<10 dpp; first blue dot, Fig. 6b; $n = 1$), 'demethylated' (13–16 dpp; orange dots, Fig. 6b; $n = 3$) and 'remethylated' (>16 dpp; subsequent blue dots, Fig. 6b; $n = 12$). A two-sample t-test with unequal variance was performed on CpG methylation values for $n = 15$ individual samples (i.e., demethylated v remethylated).

For high-coverage whole-genome bisulfite sequencing, raw reads were trimmed using Trim Galore! (v0.6.7; https://github.com/FelixKrueger/TrimGalore) by removing adapters, 10 bp from the 5′ end of all reads and removing low-quality base calls (Phred score <20). Trimmed reads were mapped to the possum genome reference GCA_011100635.1, deduplicated, and methylation calls in the CG dinucleotide context were extracted using Bismark v0.22.3[109] with the -pbat option. The resulting BAM files were imported into SeqMonk (v1.48.0; https://www.bioinformatics.babraham.ac.uk/projects/seqmonk/), 13 dpp and 16 dpp germ cell samples were grouped into a replicate set, and probes were made over features for ASM sites (Supplementary Data 3) and *H19* (Fig. S5b) with removal of duplicates. The feature probes were then quantitated using the Difference Quantitation method with minimum read count = 1, and % methylation data was extracted from the annotated probe report.

**Amplicon sequencing analysis.** Sequencing reads were imported into Geneious v2022.2.2 (https://www.geneious.com) and trimmed using the BBDuk plugin. Trimmed reads were then mapped to the 'in silico' PCR product, generated by extracting the corresponding sequence from the possum genome reference GCA_011100635.1; see Supplementary Table 5 for genomic coordinates for amplicons). The 'Find Variations/SNPs' feature within Geneious was then used to determine the nucleotide sequence(s) at the SNP site of interest (Supplementary Table 5), with a minimum variant frequency of 0.2. The coverage (total number of reads), reference raw frequency and variant raw frequency were extracted for each SNP site per amplicon. This information was used to calculate the variant frequency for each sample (Supplementary Data 4).

### Reporting summary

Further information on research design is available in the Nature Portfolio Reporting Summary linked to this article.

## Data availability

The genome assembly data generated in this study has been deposited in the NCBI BioProject database under accession: PRJNA562248. The RNA-sequencing and bisulfite sequencing data generated in this study has been deposited to the NCBI Gene Expression Omnibus (GEO) database under accession: GSE218695 (NCBI BioProject: PRJNA904814) and GSE218734 (NCBI BioProject: PRJNA905369), respectively. The raw data for the mitochondrial DNA libraries and nanopore libraries generated in this study has been deposited in the NCBI Short-Read Archive under accession: PRJNA904809. The possum reference genome and annotation used in this study are available in the NCBI database under accession: GCA_011100635.1. The possum reference mitochondrial genome used in this study is available in the NCBI database under accession: NC_003039.1. Various reference samples used in this study are in the NCBI database under accession: PRJNA623153, PRJNA323970, PRJNA525264 and PRJNA587034—see Supplementary Data 5 for further details. Consensus sequences of assembled mitochondrial genomes used in this study are given in Supplementary Data 7. The raw amplicon sequencing data generated in this study has been deposited to Github[110]. Source data are provided with this paper as a Source Data file.

## Code availability

The source code of the analysis is publicly available on GitHub[98,110].

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

## Acknowledgements

The authors are indebted to Sol Wogan, the Otago Peninsula Biodiversity Group, Josh Bradfield and Steve McDowall for their help in securing possum samples. Aaron Jeffs and Monika Zavodna of the Otago Genomics Facility are thanked for their excellent sequencing service. The authors acknowledge the facilities, scientific and

technical assistance from staff at the Otago Micro and Nanoscale Imaging (OMNI) Flow Cytometry Unit at the University of Otago as well as the New Zealand eScience Infrastructure (NeSI) high-performance computing facilities for consulting support and/or training services as part of this research. Andrew Veale and anonymous reviewers are further thanked for critically reading the manuscript. This work was funded by the University of Otago (T.A.H.), the Ministry for Business and Innovation Smart Ideas Grant (UOOX1909, T.A.H.), Predator Free 2050 Ltd (SS2/01/01, T.A.H., M.K.L. and "Jobs for Nature" A.A.), and the Biological Heritage National Science Challenge (BH3712227, NJG). The work of F.T-N was supported by the National Centre for Biotechnology Information of the National Library of Medicine (NLM), National Institutes of Health.

## Author contributions

T.A.H. and N.J.G. conceived the study. B.H., J.M., M.D., J.C., K.H., Y.G., T.H., F.T.N., N.C.L., P.D.W., O.F. and E.D.J. performed genome sequencing and/or annotation. K.S.R., B.K., B.C.R., Y.v.H., A.L.A. and W.-S.C. provided samples, meta information and/or managed breeding facilities. D.M.B. led the analysis of Fig. 2, with contributions from T.A.H. A.A. led the analysis for Fig. 3, with contributions from D.M.B. and O.O.-R. O.O.-R. led the analysis for Fig. 4, with contributions from D.M.B. D.M.B. led the analysis for Fig. 5, with contributions from F.C.B.R. and B.E.M.-W. M.K.L. led the analysis for Fig. 6 with contributions from D.M.B. and O.O.-R. D.M.B. managed data and figure formatting/compilation, while T.A.H. prepared Fig. 1, wrote the manuscript and directed the project.

## Competing interests

T.A.H. and D.M.B. are directors and shareholders of TOTOGEN Ltd, an agricultural genetics consultancy. All other authors have no conflicts of interest to declare.

## Additional information

[1]Department of Anatomy, University of Otago, Dunedin, New Zealand. [2]Faculty of Environmental Earth Science, Hokkaido University, Sapporo, Hokkaido 060-0808, Japan. [3]Department of Zoology, University of Otago, Dunedin, New Zealand. [4]School of Life and Environmental Science, Faculty of Science, The University of Sydney, Sydney, NSW, Australia. [5]Vertebrate Genome Laboratory, The Rockefeller University, New York, NY, USA. [6]Leibniz Institute for Zoo and Wildlife Research, Berlin, DE, Germany. [7]Tree of Life, Wellcome Sanger Institute, Hinxton, Cambridge, UK. [8]Graduate School of Information Science, Hyogo University, Hyogo, Japan. [9]Cognitive Genomics Research Group, Exploratory Research Center on Life and Living Systems (ExCELLS), National Institutes of Natural Sciences, Aichi, Japan. [10]Department of System Neuroscience, National Institute for Physiological Sciences, Aichi, Japan. [11]National Center for Biotechnology Information, National Library of Medicine, National Institutes of Health, Bethesda, MD, USA. [12]School of Biotechnology and Biomolecular Science, Faculty of Science, UNSW Sydney, Sydney, NSW 2052, Australia. [13]Laboratory of Neurogenetics of Language, The Rockefeller University, New York, NY 10065, USA. [14]Howard Hughes Medical Institute, Chevy Chase, MD 20815, USA. [15]Present address: Biology Department, University of Montana Western, Dillon, MT 59725, USA. [16]Present address: Health and Biosecurity, CSIRO, Canberra, ACT, Australia. [17]These authors contributed equally: Donna M. Bond, Oscar Ortega-Recalde, Melanie K. Laird. ✉e-mail: tim.hore@otago.ac.nz

