## [Peer Review File · Nature Communications]

REVIEWER COMMENTS

Reviewer #1 (Remarks to the Author):

Bond et al. have produced a near complete brushtail possum genome for the first time. They have used RNAseq data from over 100 samples to study the transition from pouch young to independence, highlighting novel communication genes. They have also used their dataset to investigate genomic imprinting in brushtail possum for the first time, identifying four genes that have not been reported in any other species.

Their RNAseq data has also enabled them to perform a lineage tracing analysis on possums in southern New Zealand, showing them to be an admix of distinct populations from Tasmania and mainland Australia.

This is an interesting study that sheds further light on the evolution of genomic imprinting. It highlights the need for de novo analysis to identify parent-of-origin specific gene expression in more distantly related species, rather than just looking at genes known to be imprinted in eutherians.

Comments to the author.

This is an interesting paper, but it lacks cohesion. The story jumps around with no flow. The section on lineage tracing whilst interesting feels out of place (especially in the middle of the paper) and should be removed and form a starting point for a population study paper.

The section on transition to life out of the pouch is compelling . However, it was not immediately clear that the data plotted on Figure 2a were average RPKM for the different paralogues. This assumes that all the paralogues are regulated in the same way. The authors should instead plot each gene separately in supplementary material to show each behaves the same and make it clear that this figure shows the paralogue average. They should also plot the galactase/CYP expression ratio as a graph if they are going to correlate other genes' expression with it.

"H19 is thought to orchestrate paternal-specific expression of insulin-like growth factor 2 (IGF2) in mice" the reference given for this statement only looks at human data. Leighton PA, et al. 1995. Nature 375:34–39 should be used instead.

The data on the changes of H19, Vulpeculin and ganderin expression between PY and adult is noteworthy but I would like to have seen the data for other genes that follow the same signature. If the authors perform a differential analysis between PY and adults they would be able to identify the genes involved in this critical transition period. They could then cluster the genes to give a more global view of what is going on.

The imprinting analysis is the most robust part of this study. Using differential methylation as a starting point is sensible but I think they should also run an allele specific expression analysis in tandem. This would allow them to identify other candidate imprinted genes. Imprinting control regions in eutherians can influence gene expression over megabases and the cutoff of 20kb they uses would miss many of the candidates.

They should also comment on why they did not identify the H19-DMD in this analysis.

They have discovered two putative novel maternally methylated DMRs that are located over promoters of three paternally expressed genes. The majority of eutherian ICRs (23) share this pattern with methylation on the maternal promoter of a paternally expressed gene. What is more noteworthy is the discovery of a putative paternally methylated intronic DMR. Only three paternally methylated ICR are know in eutherians and each of those are intergenic. The authors should comment on this in the discussion.

Reviewer #2 (Remarks to the Author):

In the manuscript entitled "The admixed genome of brushtail possum reveals invasion history in New Zealand and novel marsupial imprinted genes", the first near-complete sequence of a brushtail possum genome was produced using nanopore sequencing technology. Combined with multiple transcriptome data, the authors revealed following three main findings. (i) Some metabolic and chemo-sensory genes such as GAL, CYP, Vulpeculin and Ganderin showed striking gene expression change between pre- and post-weaning stages. (ii) Southern New Zealand possums showed high levels of genetic diversity driven by admixture between diverse sources in Tasmania, NSW and unidentified source. (iii) Four genes, GPX7, EPM2AIP1, MLH1 and UBP1 were appeared to be imprinted in brushtail possums and three potential imprinted control regions (one paternal and two maternal differentially methylated regions, DMRs) were found from these gene loci.

Identification of novel marsupial imprinted genes and DMRs by unbiased method is important to advance understanding how differently imprinting has been evolved in another mammalian lineage than eutherians. It is also notable that the parental-conflict hypothesis seems to be applied for GPX7, EPM2AIP1 and UBP1. These findings would be a significant impact on the research field of genomic imprinting because most works on marsupial genomic imprinting have been restricted to orthologues of eutherian imprinted genes for a long time.

The text in the manuscript is clear and easy to read and the data are high quality. There is no critical flaw for the data interpretation and conclusions. Below are my suggestions that could help strengthen the work.

#1 In figure legend for Fig.1, (b) is appeared twice. The second (b) should be (c) and c) should be (d).

#2 In Fig.1d, there is a description for bisulfite-sequencing of 13 individuals, 2 tissues, but it is unclear for me where these data were used in this research.

#3 For Fig.S2a, there is a description in the text "we saw high expression in possum pouch young, but not juvenile or adults". I cannot agree with this description because I see that AFP expression levels of most juveniles are still high in Fig.S2a.

#4 For Fig.2a, the text says that GAL were high in pouch young but low in adults and juveniles. But most dots in juvenile stage show similar expression levels as pouch young stage in Fig.2a. Also, the text says that CYP became highly expressed in backriders and adults. In the graph, the data from adult certainly show higher expression of CYP, but most data in juvenile stage show similar low expression levels as seen in pouch young stage.

#5 Also for Fig.2c, the text says that IGF2 was biallelically expressed in adults and juveniles. In the graph, I see that dots for adult samples are certainly close to the 50% line, but the data for juveniles are similar as that of pouch young stage.

#6 Fig.2c shows allelic ratio of IGF2 expression and it becomes biallelically expressed in adult stage. How about the change of IGF2 expression level? Is expression level increased upon the gain of biallelic expression?

#7 For the novel 4 marsupial imprinted genes and 3 DMRs, imprinted expression and differential methylation are marsupial specific? I suggest confirming it because these data can be easily obtained from public transcriptome and DNA methylation data in mouse and human. For monotremes, it might be difficult to determine their allelic expression, but checking methylation status of the orthologous CpG islands for the 3 DMRs will not be difficult if genomic DNA of a monotreme species is available. There is a hypothesis that emergence of DMRs is associated with TE insertions. Is there any marsupial-specific TE insertion event nearby the 3 DMRs which might be associated with the acquisition of differential methylation?

#8 There are allelic expression data of brain in Fig.5b, c, d, but only not in Fig.5a. Is this just a

mistake? If there is any reason, please provide an explanation.

#9 In figure legend for Fig.6, "c" should be "(c)".

#10 In Fig.6c, these 6 DMRs are hypomethylated in PGCs, nearly 50% methylated in somatic cells and hypo- or fully methylated in adult sperm. Is this methylation pattern characteristic only for DMRs associated with imprinting? I suggest checking methylation levels of the 173 ASM sites (remaining 167 sites?) in PGCs, somatic cells and adult sperm as the authors could possibly detect more DMRs associated with imprinting from the remaining ASM sites.

#11 In the third paragraph in the section entitled "Global methylation erasure...", there is a sentence "however, H19 and GPX7, were fully methylated (94.24-95.79%) (Fig. 5c)". As there is no methylation data in Fig.5c, (Fig. 5c) must be (Fig. 6c). In addition, there is a description "Supplementary Fig. 4a-f;", but I cannot find Fig.S4f.

#12 In discussion, the authors state that MLH1 does not have an obvious link to the parental conflict hypothesis. For the reason of this, it can be thought that imprinting of MLH1 is by-product of EPM2AIP1 imprinting. Because these two genes share promoter region, selective pressure for the acquisition of EPM2AIP1 imprinting might affect MLH1 expression coincidentally.

#13 In discussion, there is a sentence "Our analysis is almost certainly conservative – we have only tested 4 of the 173 sites with ASM...". It is confusing because I believe that the authors tested 173 sites and found 3 novel DMRs.

#14 Previous studies have found out the tendency that paternal DMRs are located at non-promoter regions and have relatively lower CpG density while maternal DMRs are located at promoter regions and have higher CpG density. It is interesting that this tendency is completely applied to the three DMRs discovered in this study. How about mentioning this somewhere in discussion?

Reviewer #3 (Remarks to the Author):

In this manuscript, Bond et al. sequenced and assembled the brushtail possum (*Trichosurus vulpecula*) genome. The authors combined PacBio long-read data, 10x Genomics linked-read data, as well as Hi-C and bionano data to achieve super high quality and continuity. The authors analyzed transcriptome data across developmental stages and identified gene expression signature of weaning in brushtail possum. Through SNPs called from RNAseq data, they are able to show the genetic admixture of Dunedin possums. The authors also searched for novel imprinted genes in heterozygous individuals using allele-specific methylation profiling, and allele-specific gene expression in the RNA-seq data.

Major comments:

Page 2, the authors mentioned "however, unbiased searches for marsupial imprinted genes are limited." It might be true at the time of submission, but a recent survey of marsupial imprinted genes in *Monodelphis domestica* was published <https://doi.org/10.1093/molbev/msad022>. The authors may want to compare their results with *Monodelphis*, or at least cite the paper.

The evidence for monoallelic expression for EPM2AIP1 and MLH1 is strong. However, variability was observed for GPX7 and UBP1, with some individuals showing various degree of biallelic expression (120320_MR_SDR1, 030620_PY01, 280720_PY01, 200421_MB_SDR1, 250521_MG_PF1). The authors need to discuss this in the manuscript.

Monoallelic expression can be due to genomic imprinting, random monoallelic expression, or strong cis-eQTL effects (<https://doi.org/10.1038/hdy.2014.18>). To exclude the possibility of random monoallelic expression and confirm imprinting, ideally, the parental transmission direction can be validated in reference-allele-expressing individuals and alternative-allele-expressing individuals. For IGR2R, GPX7, and EPM2AIP1, the authors were able to confirm the parental

transmission direction for samples expressing the reference and the alternative alleles, providing strong evidence of parent-of-origin effect. However, for UBP1 and MLH1, only one direction was confirmed and the a large proportion (>90%) of the informative individuals lack parent-of-origin information. I would call these two genes (UBP1 and MLH1) as candidate imprinted genes because of the possibility of random monoallelic expression.

The authors need to be cautious to claim that the study is an unbiased genome-wide search of imprinted genes. "Our discovery pipeline relied on first identifying allele specific methylation" implies a hypothesis that all imprinted genes are associated a DMR (differentially methylated region). However, in eutherian mammals, only a third of the imprinted genes have a DMR. Therefore, discovery based on ASM will miss the imprinted genes regulated by other epigenetic mechanisms, such as histone modifications.

Specific comments:

- 1) I would appreciate if the authors can add line numbers in the manuscript.
- 2) Page 5, "chromosome assembly", this assembly has very high continuity and completeness. The authors should include the genome assessment stats (contig N50, scaffold N50 etc) in the results to highlight the strength of the manuscript.
- 3) ASM discovery: what proportion of the genome was covered by >10x depth of Nanopore reads?
- 4) Figure 5, please add the total RNA-seq read depth to each SNP/tissue.
- 5) Page 12, global methylation dynamics: bisulfite sequencing may subject to low library complexity tissue, especially when the coverage is low. Please state the number of replicates clearly in the methods or results. Also, if the authors would like to claim a dramatic demethylation and remethylation, some sort of formal statistical analysis is needed. Simply comparing the overall methylation percentages is not sufficient.

RESPONSE TO REVIEWERS' COMMENTS

Reviewer #1 (Remarks to the Author):

Bond et al. have produced a near complete brushtail possum genome for the first time. They have used RNAseq data from over 100 samples to study the transition from pouch young to independence, highlighting novel communication genes. They have also used their dataset to investigate genomic imprinting in brushtail possum for the first time, identifying four genes that have not been reported in any other species.

Their RNAseq data has also enabled them to perform a lineage tracing analysis on possums in southern New Zealand, showing them to be an admix of distinct populations from Tasmania and mainland Australia.

This is an interesting study that sheds further light on the evolution of genomic imprinting. It highlights the need for *de novo* analysis to identify parent-of-origin specific gene expression in more distantly related species, rather than just looking at genes known to be imprinted in eutherians.

We thank reviewer 1 for favourably identifying the novel aspects of our work, and its range of impact from population genetics to developmental biology and epigenetics. We also agree that our *de novo* search for imprinted genes is both timely and interesting, as well as providing much needed insight into the evolution of imprinting.

Comments to the author.

This is an interesting paper, but it lacks cohesion. The story jumps around with no flow. The section on lineage tracing whilst interesting feels out of place (especially in the middle of the paper) and should be removed and form a starting point for a population study paper.

Following the direction from the editor, we have retained the lineage tracing section in this manuscript. While lineage tracing is an important part of the population genetics story, in our opinion, the admixture aspect feeds nicely into genomic imprinting studies, because admixture provides greater imprinted gene search resolution.

The section on transition to life out of the pouch is compelling . However, it was not immediately clear that the data plotted on Figure 2a were average RPKM for the different paralogues. This assumes that all the paralogues are regulated in the same way. The authors should instead plot each gene separately in supplementary material to show each behaves the same and make it clear that this figure shows the paralogue average. They should also plot the galactase/CYP expression ratio as a graph if they are going to correlate other genes' expression with it.

To address this point we have provided the *GAL/CYP* ratio as a line plot in the main text (Fig. 2a), and made our axis labels and figure legend clearer regarding the nature of this ratio (i.e. the fact it is an average of multiple paralogues rather than representing individual genes).

In addition, the expression levels (RPKM - log2 values) for each *GAL* and *CYP* paralogue used for this analysis is given in Supplementary Table 3. We have also added plots with the expression profiles for these individual paralogues to this file.

"H19 is thought to orchestrate paternal-specific expression of insulin-like growth factor 2 (IGF2) in mice" the reference given for this statement only looks at human data. Leighton PA, et al. 1995. Nature 375:34–39 should be used instead.

We have now corrected this error.

The data on the changes of H19, Vulpeculin and ganderin expression between PY and adult is noteworthy but I would like to have seen the data for other genes that follow the same signature. If the authors perform a differential analysis between PY and adults they would be able to identify the genes involved in this critical transition period. They could then cluster the genes to give a more global view of what is going on.

To be clear about our analysis, we initially found the *GAL/CYP* ratio by comparing PY and adults, and then we searched for other genes that showed co-regulation with this change in liver metabolism at weaning. In doing so, we uncovered *H19*, *vulpeculin* and *ganderin* as these show the greatest correlation (or anti-correlation) to the *GAL/CYP* ratio - i.e., the metabolic weaning 'signature' we uncovered. We have since updated the manuscript to be clearer about how we found the *GAL/CYP* ratio (i.e. differential analysis between PY and adults):

"Perhaps the most striking gene expression change we observed in liver between pouch young and adults was associated with the metabolic effects of weaning (Fig. 2a)."

Nevertheless, we now realise that readers could be interested in weaning associated genes in addition to those we have discussed. For this reason we have supplied the expression levels (RPKM - log2 values) for all genes, and sorted them based on their correlation to the *GAL/GYP* ratio. This data is supplied in Supplementary Table 3.

The imprinting analysis is the most robust part of this study. Using differential methylation as a starting point is sensible but I think they should also run an allele specific expression analysis in tandem. This would allow them to identify other candidate imprinted genes. Imprinting control regions in eutherians can influence gene expression over megabases and the cutoff of 20kb they uses would miss many of the candidates.

While this was not detailed in the original manuscript, previously we have searched for imprinted genes in possums using only allele-specific expression analysis (Reese, 2021, Thesis). We were not satisfied with this approach as most candidate genes were false positives, often caused by non-expressed pseudogene DNA sequences providing the false impression of an imprinted expression in its paralogue. Our strike-rate identifying imprinted genes improved greatly once we used allele specific methylation identified via nanopore sequencing. This was the primary mechanism to identify candidate imprinted regions as we searched for mono-allelic expression nearby.

In the initial submission, we did this in a rather *ad hoc* manual manner (focusing on genes next to ASM sites), however, during revision we performed a search for genome-wide mono-allelic

expression of SNPs in Sandy's brain, liver and muscle, and then overlapped this with our ASMs using a +/- 1-Mbp cut off - the average size of the largest imprinted domains in mammals (MacDonald and Mann, 2020, PMID: 32760061; Hubert and Demars, 2022, PMID: 35368671). This resulted in a list of 91, 61 and 62 genes with potential mono-allelic expression in RNA-sequencing data. The list was then filtered based on evidence of biallelic expression of additional SNPs in the genes of interest and/or removal of false positives. As a final 'sanity-check' for the candidate imprinted regions passing these filters, the associated ASMs were examined in germ cells undergoing epigenetic reprogramming. Of these, we found 8 ASMs showing the expected methylation pattern for imprinting (unmethylated in germ cells, around ~50% in somatic cells, either ~100% or ~0% methylation in adult sperm) (Fig 6d, lower panel).

These high confidence candidates will ultimately need comprehensive verification in subsequent studies, but provides a useful starting point for further analysis and the opportunity to compare with other studies (e.g. opossum - see below).

They should also comment on why they did not identify the H19-DMD in this analysis.

The animal that formed the basis of our genome sequence (Sandy) was homozygous at the *H19* locus, hence it was not detected in the ASM analysis (which relies on heterozygous SNPs to generate the methylation haplotypes, or allele specific methylation patterns). We have now added an explanation of this in the discussion of our manuscript.

They have discovered two putative novel maternally methylated DMRs that are located over promoters of three paternally expressed genes. The majority of eutherian ICRs (23) share this pattern with methylation on the maternal promoter of a paternally expressed gene. What is more noteworthy is the discovery of a putative paternally methylated intronic DMR. Only three paternally methylated ICR are known in eutherians and each of those are intergenic. The authors should comment on this in the discussion.

Reviewer 1 makes an interesting observation here that we had overlooked. This has now been added to the discussion following their suggestion:

"Associated with these genes was parent-specific methylation - two instances where the maternal chromosome was methylated (like most imprinted loci), and one example of paternal methylation (GPX7). Interestingly, this site of differential methylation is located in an intron, similar to other known paternally methylated imprint control regions, and unlike maternal imprint control regions which are generally located at transcription start sites⁷⁰."

Reviewer #2 (Remarks to the Author):

In the manuscript entitled "The admixed genome of brushtail possum reveals invasion history in New Zealand and novel marsupial imprinted genes", the first near-complete sequence of a brushtail possum genome was produced using nanopore sequencing technology. Combined with multiple transcriptome data, the authors revealed following three main findings. (i) Some metabolic and chemo-sensory genes such as GAL, CYP, Vulpeculin and Ganderin showed striking gene expression change between pre- and post-weaning stages. (ii) Southern New Zealand possums showed high levels of genetic diversity driven by admixture between diverse sources in Tasmania, NSW and unidentified source. (iii) Four genes, GPX7, EPM2AIP1, MLH1 and UBP1 were appeared to be imprinted in brushtail possums and three potential imprinted control regions (one paternal and two maternal differentially methylated regions, DMRs) were found from these gene loci.

Identification of novel marsupial imprinted genes and DMRs by unbiased method is important to advance understanding how differently imprinting has been evolved in another mammalian lineage than eutherians. It is also notable that the parental-conflict hypothesis seems to be applied for GPX7, EPM2AIP1 and UBP1. These findings would be a significant impact on the research field of genomic imprinting because most works on marsupial genomic imprinting have been restricted to orthologues of eutherian imprinted genes for a long time.

The text in the manuscript is clear and easy to read and the data are high quality. There is no critical flaw for the data interpretation and conclusions. Below are my suggestions that could help strengthen the work.

#1 In figure legend for Fig.1, (b) is appeared twice. The second (b) should be (c) and c) should be (d).

We have now corrected this error.

#2 In Fig.1d, there is a description for bisulfite-sequencing of 13 individuals, 2 tissues, but it is unclear for me where these data were used in this research.

The bisulfite sequencing data was used in the epigenetic reprogramming studies. We have now clarified this in the Fig. 1 legend as follows:

"A range of functional genomics data supports the assembly, including: RNA-sequencing (mostly liver) from a wide population of possums and 13 tissues from Sandy. Also performed was DNA methylation profiling, featuring nanopore and bisulfite sequencing for genomic imprinting and germline reprogramming analysis, respectively, and a sequence assembly of a Tasmanian possum."

#3 For Fig.S2a, there is a description in the text "we saw high expression in possum pouch young, but not juvenile or adults". I cannot agree with this description because I see that AFP expression levels of most juveniles are still high in Fig.S2a.

This was an oversight on our behalf. We were not clear that juveniles are actually pouch young >120 days old (for which we were not able to accurately date). As such we have corrected this sentence to the following:

“Yet we saw high expression in possum pouch young, including juvenile pouch young >120 d post birth (Fig. S2a).”

We have also updated our methods to be clearer about how developmental staging was undertaken, and what the terms ‘juvenile’ and ‘backrider’ actually mean:

“Pouch young >120 days of development were challenging to stage, and for this reason have been referred to as ‘juveniles’ (i.e. pouch-young greater than >120 d, but younger than ~180 days, upon which time they leave the pouch). The sole ‘back-rider’ was Sheila, who was captured with her mother (Puku) soon after having left the pouch.”

#4 For Fig.2a, the text says that GAL were high in pouch young but low in adults and juveniles. But most dots in juvenile stage show similar expression levels as pouch young stage in Fig.2a. Also, the text says that CYP became highly expressed in backriders and adults. In the graph, the data from adult certainly show higher expression of CYP, but most data in juvenile stage show similar low expression levels as seen in pouch young stage.

We have corrected this sentence to be congruent with the pouch-young, juvenile and adult definitions as for the comment above:

“Genes associated with the Leloir pathway, involved with the catabolism of galactose and other carbohydrates, were high in pouch young, but low in adults (Supplementary Table 3).”

#5 Also for Fig.2c, the text says that IGF2 was biallelically expressed in adults and juveniles. In the graph, I see that dots for adult samples are certainly close to the 50% line, but the data for juveniles are similar as that of pouch young stage.

We have corrected this sentence to be congruent with the pouch-young, juvenile and adult definitions as for the comment above:

“We found exclusive maternal expression of H19, whereas IGF2 was paternally expressed in pouch young and biallelically expressed in adults where H19 expression was silenced (Fig. 2b,c; Supplementary Fig. S2d; Supplementary Table 4).”

#6 Fig.2c shows allelic ratio of IGF2 expression and it becomes biallelically expressed in adult stage. How about the change of IGF2 expression level? Is expression level increased upon the gain of biallelic expression?

We found that *IGF2* expression decreased during development/aging (see figure below), a result consistent with our understanding of the insulin like growth factors in human - *IGF2* is predominantly a fetal growth factor that gets silenced during early postnatal life in humans, with its expression replaced by *IGF1*.

IGF2 in possum

IGF1 in possum

We feel this result does not justify a new figure in the manuscript, however, we could provide one if the reviewer/editor believes we should.

#7 For the novel 4 marsupial imprinted genes and 3 DMRs, imprinted expression and differential methylation are marsupial specific? I suggest confirming it because these data can be easily obtained from public transcriptome and DNA methylation data in mouse and human. For monotremes, it might be difficult to determine their allelic expression, but checking methylation status of the orthologous CpG islands for the 3 DMRs will not be difficult if genomic DNA of a monotreme species is available. There is a hypothesis that emergence of DMRs is associated with TE insertions. Is there any marsupial-specific TE insertion event nearby the 3 DMRs which might be associated with the acquisition of differential methylation?

This is an interesting question which we are actively trying to address. *MLH1/EPM2AIP1* expression and methylation has been extensively studied in humans (Lynch et al., 2015), and is not imprinted. Nevertheless, there are indications from cattle that parent-of-origin effects might exist at this locus (Kenny et al., 2022; PMID: 35910233), implying that at least imprinting of *MLH1/EPM2AIP1* is not marsupial specific. Experiments are underway to definitively test *MLH1/EPM2AIP1* imprinting (and that of *UBP1* and *GPX7*) in other eutherian, marsupial and monotreme species using tissue samples collected for this purpose, as well as publicly available datasets. While this work is exciting, it is ongoing, and we believe beyond the scope of this manuscript.

#8 There are allelic expression data of brain in Fig.5b, c, d, but only not in Fig.5a. Is this just a mistake? If there is any reason, please provide an explanation.

Figure 5a shows mono-allelic expression of *GPX7* in 'Sandy' the individual sequenced in the genome. The number of RNA-seq reads for brain is <5, so it is not presented (as stated in the figure legend).

#9 In figure legend for Fig.6, "c)" should be "(c)".

This has been corrected.

#10 In Fig.6c, these 6 DMRs are hypomethylated in PGCs, nearly 50% methylated in somatic cells and hypo- or fully methylated in adult sperm. Is this methylation pattern characteristic only for DMRs associated with imprinting? I suggest checking methylation levels of the 173 ASM sites (remaining 167 sites?) in PGCs, somatic cells and adult sperm as the authors could possibly detect more DMRs associated with imprinting from the remaining ASM sites.

We have looked at the 173 ASM sites as suggested. We have added a heatmap to Fig. 6 (i.e. Fig. 6d - shown above). The top panel shows methylation in PGCs, sperm and somatic cells for all ASM sites (excluding those with missing data, n=32) sorted by somatic methylation. We identified 41 sites that have somatic methylation between 30 and 57% (i.e. the approximate range of somatic methylation exhibited by sites with confirmed mono-allelic expression in Fig. 6c). Of these 41 sites, 32 (including the 6 ASMs presented in Fig. 6c) display methylation for PGCs and sperm within acceptable ranges to suggest imprinting (<15% for PGCs, <5% or >94% for sperm; see Supplementary Table 6). The bottom panel of Fig. 6D shows the ASM sites that are associated with genes that have confirmed imprinted, mono-allelic expression or are candidate imprinted genes (n = 13).

#11 In the third paragraph in the section entitled "Global methylation erasure...", there is a sentence "however, H19 and GPX7, were fully methylated (94.24-95.79%) (Fig. 5c)". As there is no methylation data in Fig.5c, (Fig. 5c) must be (Fig. 6c). In addition, there is a description "Supplementary Fig. 4a-f;" , but I cannot find Fig.S4f.

These were typographical oversights which we regret, and have now been corrected.

#12 In discussion, the authors state that MLH1 does not have an obvious link to the parental conflict hypothesis. For the reason of this, it can be thought that imprinting of MLH1 is by-product of EPM2AIP1 imprinting. Because these two genes share promoter region, selective pressure for the acquisition of EPM2AIP1 imprinting might affect MLH1 expression coincidentally.

This possibility that *MLH1* is a bystander of *EPM2AIP1* imprinting is a pertinent point, which we have now made clearer in the discussion:

“The MLH1 gene does not have an obvious link to the parental conflict hypothesis, and therefore may be a ‘bystander’ caught in the imprinted regulation of closely linked EPM2AIP1.”

#13 In discussion, there is a sentence “Our analysis is almost certainly conservative – we have only tested 4 of the 173 sites with ASM...”. It is confusing because I believe that the authors tested 173 sites and found 3 novel DMRs.

We apologise for not being clearer here - of the 173 sites identified as having allele-specific methylation (i.e. ‘ASMs’), we have only comprehensively tested 3 sites for parent-specific expression in neighbouring genes. This is detailed in the revised manuscript as follows:

“It is likely more imprinted genes in possum will be uncovered - of the 173 ASM sites identified by nanopore sequencing, we have performed experiments validating parent-specific expression on genes associated with just 3 sites.”

#14 Previous studies have found out the tendency that paternal DMRs are located at non-promoter regions and have relatively lower CpG density while maternal DMRs are located at promoter regions and have higher CpG density. It is interesting that this tendency is completely applied to the three DMRs discovered in this study. How about mentioning this somewhere in discussion?

As for Reviewer 1, Reviewer 2 makes an interesting observation here that we had overlooked. This has now been added to the discussion following their suggestion:

“Associated with these genes was parent-specific methylation - two instances where the maternal chromosome was methylated (like most imprinted loci), and one example of paternal methylation (GPX7). Interestingly, this site of differential methylation is located in an intron, similar to other known paternally methylated imprint control regions, and unlike maternal imprint control regions which are generally located at transcription start sites⁷⁰.”

Reviewer #3 (Remarks to the Author):

In this manuscript, Bond et al. sequenced and assembled the brushtail possum (*Trichosurus vulpecula*) genome. The authors combined PacBio long-read data, 10x Genomics linked-read data, as well as Hi-C and bionano data to achieve super high quality and continuity. The authors analyzed transcriptome data across developmental stages and identified gene expression signature of weaning in brushtail possum. Through SNPs called from RNAseq data, they are able to show the genetic admixture of Dunedin possums. The authors also searched for novel imprinted genes in heterozygous individuals using allele-specific methylation profiling, and allele-specific gene expression in the RNA-seq data.

Major comments:

Page 2, the authors mentioned “however, unbiased searches for marsupial imprinted genes are limited.” It might be true at the time of submission, but a recent survey of marsupial imprinted genes in *Monodelphis domestica* was published <https://doi.org/10.1093/molbev/msad022>. The authors may want to compare their results with *Monodelphis*, or at least cite the paper.

The reviewer is correct - the mentioned *Monodelphis domestica* paper came out during the manuscript review process. We have now cited this work in the introduction and results section (Fig. 6d), and also described some of their results in our discussion:

“In addition, a recently-published analysis of RNA-sequencing in Monodelphis domestica estimates there may be up to 60 imprinted genes in the American opossum¹⁷.”

The evidence for monoallelic expression for EPM2AIP1 and MLH1 is strong. However, variability was observed for GPX7 and UBP1, with some individuals showing various degree of biallelic expression (120320_MR_SDR1, 030620_PY01, 280720_PY01, 200421_MB_SDR1, 250521_MG_PF1). The authors need to discuss this in the manuscript.

We have changed our description of UBP1 and GPX7 expression to include ‘bias’ rather than strict mono-allelic expression. For example, the results now reads:
While monoallelic expression was clear for MLH1/EPM2AIP1, there appeared to be biased expression of GPX7 and UBP1.

And the discussion now reads:

Using long-read methylation sequencing as a starting point, we discovered 4 genes with parent-specific imprinted expression across a wide range of tissues, albeit with two genes (UBP1 and GPX7) showing some evidence for leaky monoallelic expression, as previously reported for some marsupial imprinted genes²².

Monoallelic expression can be due to genomic imprinting, random monoallelic expression, or strong cis-eQTL effects (<https://doi.org/10.1038/hdy.2014.18>). To exclude the possibility of random monoallelic expression and confirm imprinting, ideally, the parental transmission direction can be validated in reference-allele-expressing individuals and alternative-allele-expressing individuals. For IGR2R, GPX7, and EPM2AIP1, the authors were able to confirm the parental transmission direction for samples expressing the reference and the alternative alleles, providing strong evidence of parent-of-origin effect. However, for UBP1 and MLH1, only one direction was confirmed and the a

large proportion (>90%) of the informative individuals lack parent-of-origin information. I would call these two genes (*UBP1* and *MLH1*) as candidate imprinted genes because of the possibility of random monoallelic expression.

Our population analysis from heterozygous individuals clearly shows expression is monoallelic, but that both versions of the SNP can be expressed in the population. Thus, cis-eQTL can not be the explanation for monoallelic expression for our 4 novel imprinted genes (*MLH1*, *EPM2AIP1*, *GPX7* and *UBP1*).

Random monoallelic expression is common at certain gene clusters, like olfactory and immune genes, or genes subject to transcriptional 'bursting' (Eckersley-Maslin, 2014, PMID: 24780084). Importantly, random monoallelic expression features the expression of alternative alleles within the same individual and tissue (hence the term, 'random', and resulting in mosaic expression patterns, like the calico cat). For it to be confused with imprinted gene expression like we saw (with monoallelic expression in all tissues), the 'random' inactivating event would need to have occurred in the zygote or somehow affect the same allele in all of the progenitor cells giving rise to the embryo. There is to our knowledge, no example of such a non-imprinted, random monoallelically expressed gene that affects all tissues of an individual. Hence we consider this possibility very unlikely.

Nevertheless, to bolster our data, we did identify an additional duo for *MLH1* and *UBP1*, and saw the same directionality of imprinting in both cases. This data has been added to Supplementary Table 4.

The authors need to be cautious to claim that the study is an unbiased genome-wide search of imprinted genes. "Our discovery pipeline relied on first identifying allele specific methylation" implies a hypothesis that all imprinted genes are associated a DMR (differentially methylated region). However, in eutherian mammals, only a third of the imprinted genes have a DMR. Therefore, discovery based on ASM will miss the imprinted genes regulated by other epigenetic mechanisms, such as histone modifications.

We respect the reviewer's position on this point, and have now removed all references to an 'unbiased' search for imprinted genes.

Specific comments:

1) I would appreciate if the authors can add line numbers in the manuscript.

This has now been updated.

2) Page 5, "chromosome assembly", this assembly has very high continuity and completeness. The authors should include the genome assessment stats (contig N50, scaffold N50 etc) in the results to highlight the strength of the manuscript.

This has now been updated in the manuscript as follows:

This resulting assembly had 99.88% of the sequence mapped to chromosomes, matching the karyotype (Fig. 1c), and possessing strong assembly rates (scaffold N50, 442,560,073 bp; contig N50, 4,314,688 bp).

3) ASM discovery: what proportion of the genome was covered by >10x depth of Nanopore reads?

We used the tool bedcov from the package SAMtools (v1.16.1) to calculate the coverage at 5X and 10X for the nanopore reads. Using this methodology we estimated a coverage of 98% and 80.1%, respectively. We included this information in the methods of the revised manuscript.

4) Figure 5, please add the total RNA-seq read depth to each SNP/tissue.

As in the original manuscript, RNA-sequencing read depth for all SNPs analysed is given in Supplementary Table 4. To further help readers assess the strength of data, we have added the RNA-sequencing read depth for the heterozygous individuals presented in Fig. 5, Fig. S2d and Fig. S4. We have also explained that these numbers represent the read depth values (≥ 5) in the legends for these figures.

5) Page 12, global methylation dynamics: bisulfite sequencing may subject to low library complexity tissue, especially when the coverage is low. Please state the number of replicates clearly in the methods or results. Also, if the authors would like to claim a dramatic demethylation and remethylation, some sort of formal statistical analysis is needed. Simply comparing the overall methylation percentages is not sufficient.

In response to this comment, we have stated the replicates used more clearly in the methods section ("DNA methylation analysis"). We agree that a statistical comparison is appropriate, however, since we have only one sample in our dataset representing PGCs before methylation reduction (see first blue dot, Fig. 6b), we could not include this time point in a statistical test and have removed the word 'dramatically' associated with demethylation in the results. We instead performed a t-test between the average CpG methylation in the 'demethylated' samples (orange dots, Fig. 6b; $n=3$) vs 'remethylated' samples (subsequent blue dots, Fig. 6b; $n=12$) and found that methylation differed significantly ($p = 0.00002906$), as expected. We have included details of this statistical comparison in the methods and results sections.

REVIEWERS' COMMENTS

Reviewer #1 (Remarks to the Author):

This is my second time in reviewing this article. The authors have addressed most of my comments after the first review.

I am pleased that the authors are now able to include a comparison of their findings with those from Cao et al. Of particular interest is their finding that 4 of their novel ASMs are located near genes

Cao et al identified to have imprinted expression in opossum. This suggests that marsupial may have their own set of imprinted domains. This will be of interest to others in the marsupial imprinting field.

Reviewer #2 (Remarks to the Author):

The manuscript has been revised well and the responses to my comments and suggestions are acceptable.

RESPONSE TO REVIEWERS' COMMENTS

Reviewer #1 (Remarks to the Author):

This is my second time in reviewing this article. The authors have addressed most of my comments after the first review.

I am pleased that the authors are now able to include a comparison of their findings with those from Cao et al. Of particular interest is their finding that 4 of their novel ASMs are located near genes Cao et al identified to have imprinted expression in opossum. This suggests that marsupial may have their own set of imprinted domains. This will be of interest to others in the marsupial imprinting field.

We are also pleased with the outcome of our comparison with Cao et al. and agree that it will be of interest to others in the marsupial imprinting field and those interested in the evolution of this important mechanism. Thank you for your comments and taking the time to review our manuscript.

Reviewer #2 (Remarks to the Author):

The manuscript has been revised well and the responses to my comments and suggestions are acceptable.

Thank you for reviewing our manuscript. We are pleased that our responses to your comments and suggestions are acceptable.